# Pregnancy reprograms the epigenome of mammary epithelial cells and blocks the development of premalignant lesions

Mary J. Feigman[1,5], Matthew A. Moss[2,5], Chen Chen[1,5], Samantha L. Cyrill[1], Michael F. Ciccone[1], Marygrace C. Trousdell [1], Shih-Ting Yang[1], Wesley D. Frey [3], John E. Wilkinson[4] & Camila O. dos Santos [1✉]

Pregnancy causes a series of cellular and molecular changes in mammary epithelial cells (MECs) of female adults. In addition, pregnancy can also modify the predisposition of rodent and human MECs to initiate oncogenesis. Here, we investigate how pregnancy reprograms enhancer chromatin in the mammary epithelium of mice and influences the transcriptional output of the oncogenic transcription factor cMYC. We find that pregnancy induces an expansion of the active cis-regulatory landscape of MECs, which influences the activation of pregnancy-related programs during re-exposure to pregnancy hormones in vivo and in vitro. Using inducible *cMYC* overexpression, we demonstrate that post-pregnancy MECs are resistant to the downstream molecular programs induced by cMYC, a response that blunts carcinoma initiation, but does not perturb the normal pregnancy-induced epigenomic landscape. *cMYC* overexpression drives post-pregnancy MECs into a senescence-like state, and perturbations of this state increase malignant phenotypic changes. Taken together, our findings provide further insight into the cell-autonomous signals in post-pregnancy MECs that underpin the regulation of gene expression, cellular activation, and resistance to malignant development.

[1] Cold Spring Harbor Laboratory, Cold Spring Harbor, New York, NY 11724, USA. [2] Donald and Barbara Zucker School of Medicine at Hofstra/Northwell, Hempstead, NY 11549, USA. [3] School of Medicine, Tulane University, New Orleans, LA 70118, USA. [4] Department of Comparative Medicine, University of Washington, Seattle, WA 98195, USA. [5] These authors contributed equally: Mary J. Feigman, Matthew A. Moss, Chen Chen. ✉email: dossanto@cshl.edu

In mammals, the physiological stimulus brought by pregnancy results in considerable developmental reorganization, with mammary glands playing pivotal role in milk production and offspring nourishment. Substantial tissue remodeling, including the expansion of epithelial cells and ductal structures, is followed by the accumulation of milk droplets as gestation progresses. During lactation, milk production is synchronized with milk release by a series of transcriptional and mechanical events in luminal and myoepithelial cells. As lactation ceases, the mammary gland returns to a nonsecretory state and adopts a tissue organization that resembles the prepregnancy one[1].

However, post-pregnancy mammary epithelial cells (MECs) are distinct from their pre-pregnancy counterparts, with transcriptional networks being differentially regulated in post-pregnancy mammary tissue from mice/rodents and humans[2–6]. Several reports have also suggested that post-pregnancy mammary glands from several mammalian species respond robustly to the signals of consecutive pregnancies[7–10], suggesting a molecular memory of prior pregnancies. In fact, whole-genome bisulfite sequencing has revealed that pregnancy induces stable and specific changes to DNA methylation in MECs[11,12]. These epigenetic alterations correlated with the enhanced kinetics of gene reactivation in a subsequent pregnancy, and suggest that loss of DNA methylation may underlie epigenetic memory in the post-pregnancy mammary gland[12].

Pregnancy signals exert functions in the MECs that go beyond the primary function of milk production and secretion, and include modifying the risk of breast cancer in rodents and humans. While there is an increase in breast cancer risk for roughly 5–10 years after parturition[13–15], there is a long-term reduction of breast cancer risk for women completing a full-term pregnancy before the age of 30[16–19]. In several rodent systems, the inhibition of carcinogen-induced and genetic-induced mammary tumorigenesis has also been reported post pregnancy or upon pre treatment with pregnancy hormones[17,20–22]. Given the stability of the molecular programs instated by pregnancy in MECs, and the longevity of cancer preventive effects in rodents and humans, it is likely that these protective effects have a nongenetic basis.

To evaluate this hypothesis, we characterized the dynamics of gene reactivation and enhancer organization in murine MECs as they respond to pregnancy signals or early oncogenesis. Analyses of the active enhancer landscape (using H3K27ac ChIP-seq), revealed stable epigenomic alterations that influence the transcriptional output of post-pregnancy MECs in response to pregnancy signals in fat-pad transplantation assays and in organoid systems.

To characterize the influence of a pregnancy-induced epigenome on the response to oncogene expression, we used a transgenic mouse strain (CAGMYC), in which overexpression of the oncogene *cMYC*, an inducer of mammary tumor development[23], is driven in a doxycycline (DOX)-dependent manner. Post-pregnancy MECs did not undergo malignant transformation in response to *cMYC* overexpression under in vivo or in vitro conditions, in marked contrast to pre-pregnancy MECs, which engaged in abnormal, carcinoma-like growth. Transcriptomic and epigenetic analysis illustrated that *cMYC* overexpression drives post-pregnancy MECs into a senescence-like state, and perturbations to such state increased malignant phenotypic changes. Overall, our studies provided new insights into the role for pregnancy in altering epigenomic landscapes and in suppressing the malignant transformation of MECs, and suggest that the influence of pregnancy on breast cancer risk can occur, at least in part, via epigenomic reprogramming.

## Results

**Characterization of the pregnancy-induced mammary epigenome.** Our previous observation that pregnancy induces loss of DNA methylation at specific genomic regions in post-pregnancy MECs suggests that such regions assume an active regulatory state after pregnancy[12]. To test this hypothesis, we mapped global gene expression (RNA-seq) of FACS-isolated luminal MECs from nulliparous (pre-pregnancy) and parous (post-pregnancy = 21 days of gestation, 20 days of lactation, 60 days of post-lactation involution) Balb/c female mice, as well as MECs harvested from female mice during exposure to pregnancy hormones (EPH). For the first and second EPH time points, nulliparous or parous female mice, were treated with slow-released estrogen and progesterone hormones for short-term exposure (6 and 12 days) (Supplementary Fig. 1a). This procedure ensures precise timing of pregnancy-hormone exposure in nulliparous and parous female mice, and promotes mammary histological and epigenetic modifications that closely resemble those in mice exposed to pregnancy hormones following conception[12,24].

Unsupervised, global gene expression analysis of pre- and post-pregnancy luminal MECs demonstrated overall similar transcriptional programs, suggesting that a pregnancy cycle does not alter epithelial identity during tissue homeostasis (Fig. 1a, b). Focused analysis of genes correlated with MEC parity status[25] confirmed the upregulation of 38% of the parity-induced genes in post-pregnancy luminal MECs (Supplementary Fig. 1b). Luminal MECs harvested during the early stages of a second EPH (D6) clustered together with those harvested at a later time-point during the first EPH (D12), suggesting that post-pregnancy MECs activate pregnancy-induced transcription earlier in response to re-exposure to pregnancy signals (Fig. 1a, b).

To determine whether this response to re-exposure to pregnancy signals was linked to epigenetic changes, we profiled the active histone mark H3K27ac in the same cohort of luminal MECs subjected to RNA-seq. Total peak analysis revealed that pregnancy substantially expanded the active regulatory landscape of luminal MECs, with post-pregnancy MECs displaying an approximately 10-fold increase in H3K27ac peaks ($n = 207,585$), in contrast to pre-pregnant MECs ($n = 19,985$) (Fig. 1c). Regulatory regions exclusive to post-pregnancy MECs showed a 38-fold gain of H3K27ac peaks at genic regions ($n = 145,917$), and a 53-fold gain at intergenic regions ($n = 45,174$), over the same regions in pre-pregnancy MECs, suggesting that pregnancy-induced changes may expand the MEC enhancer landscape (Supplementary Fig. 1c). Gene ontology analysis (GO terms) demonstrated that pre-pregnancy H3K27ac exclusive regions were located near genes encoding protein functions associated with myeloid differentiation, cell–cell junction, and epithelial cell morphogenesis, while H3K27ac peaks exclusive to post-pregnancy MECs were enriched for pathways involved in regulation of histone H3-K27 methylation, transcription in response to UV-induced DNA damage, and regulation of gluconeogenesis (Supplementary Fig. 1d, e). These observations suggest that pre- and post-pregnancy MECs may regulate distinct molecular pathways during mammary tissue homeostasi. The H3K27ac landscape changes were also detected in luminal MECs during pregnancy, indicating that pregnancy signals are key inducers of these enhancer changes, which our analysis suggests are stably maintained in subsequent pregnancies (Fig. 1d and Supplementary Fig. 1f)

To identify the relationship between post-pregnancy H3K27ac peaks and their role in enhancer-mediated gene regulation, we used the ROSE algorithm to combine nearby peaks located at genic and intergenic regions and to delineate candidate enhancers/super enhancer regions. H3K27ac peaks exclusive to pre-pregnancy MECs define ~5000 enhancers/super enhancers, in

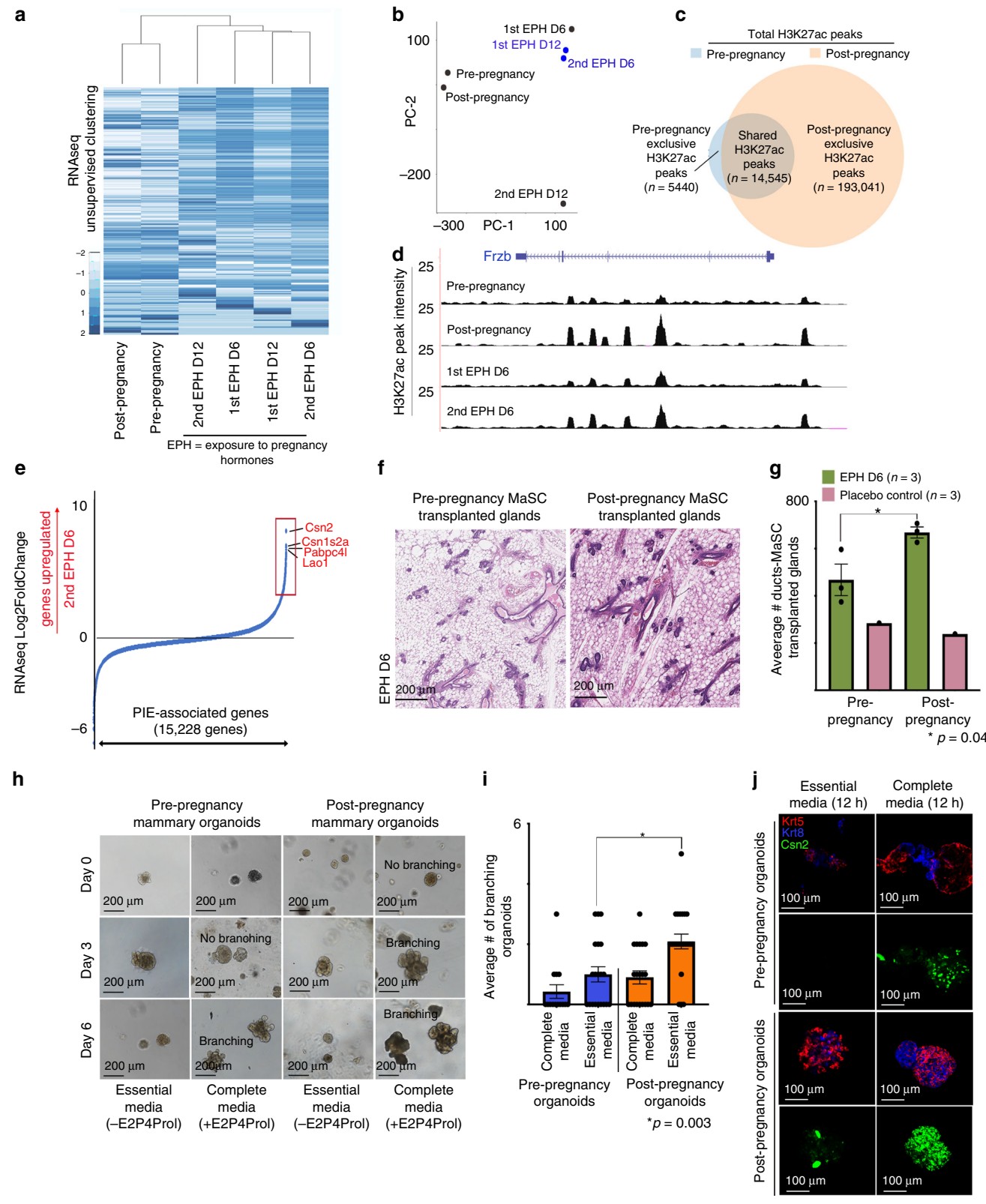

contrast to ~60,000 enhancers/super enhancer defined by H3K27ac peaks exclusive to post-pregnancy MECs (parity-induced elements, PIEs), consistent with pregnancy expanding the active enhancer landscape (Supplementary Fig. 1g). Further analysis demonstrated that most PIEs have the H3K27ac mark in MECs harvested from both first and second EPH ($n = 2263$), however more PIEs (2-fold) were active only during the second

EPH, suggesting that such elements play a role during re-exposure to pregnancy hormones (Supplementary Fig. 1h).

In addition, we identified ~15 K genes associated with PIEs, which we used to understand the effects of EPH and gene reactivation of luminal MECs. Over 600 PIE-associated genes were upregulated 16-fold or higher in luminal MECs harvested from mice during a second EPH (Log2FoldChange > 4, red box),

**Fig. 1 Characterization of the pregnancy-induced mammary epigenome. a** Heatmap distribution of gene expression data collected from FACS-isolated luminal MECs harvested from female mice at several developmental stages. **b** Principal component analysis of gene expression datasets from FACS-isolated luminal MECs harvested from female mice at several developmental stages. **c** Venn diagram demonstrating the number of shared and exclusive H3K27ac ChIP-seq peaks of FACS-isolated MECs from pre-pregnancy female mice (blue circle) and post-pregnancy female mice (orange circle). **d** Genome browser tracks showing distribution of H3K27ac peaks at distinct pregnancy cycles for Frzb locus. **e** Expression of genes associated with parity-induced elements (PIEs), according to Log2FoldChange (differential expression) in luminal MECs harvested from female mice during first and second exposure to pregnancy hormones (EPH). Boxes indicate genes upregulated during second exposure to pregnancy hormones (Log2FoldChange > 2, red). **f**, **g** H&E-stained histology images and duct quantification from mammary glands transplanted with pre-pregnancy CD1d+ MaSCs (**f**, left panel) or post-pregnancy CD1d+ MaSCs (**g**, right panel), harvested on day 6 of pregnancy-hormone exposure (EPH). $n = 3$ mammary glands injected with pre-pregnancy CD1d+ MaSCs, and $n = 3$ mammary glands injected with post-pregnancy CD1d+ MaSCs. *$p = 0.04$. Scale: 200 μm. **h**, **i** Representative images and branching quantification of mammary organoid culture derived from pre- and post-pregnancy MECs (Balb/C mice), grown with either essential media or complete media (containing estrogen (E2), progesterone (P4), and prolactin (Prol). $n = 3$ independent biological replicates. *$p = 0.02$ and **$p = 0.003$. Scale: 200 μm. **j** Immunofluorescence images of mammary organoid culture derived from pre- and post-pregnancy MECs, grown with either essential media or complete media, visualizing, KRT8 (blue), KRT5 (red), and CSN2 (green). Scale: 100 μm. For all analysis, error bars indicate standard error of mean across samples of same experimental group. $p$ values were defined using Student $t$ test.

in comparison with those cells harvested from mice exposed during first EPH (Fig. 1e). These upregulated PIE genes were enriched for functions involved in milk production[26], thus supporting that the pregnancy-induced enhancer landscape associates with activation of pregnancy-related programs in response to re-exposure to pregnancy hormones.

Our analyses of total luminal MECs do not exclude the possibility that less differentiated mammary stem cells (MaSCs) and progenitor cells could be involved in the epigenetic memory of pregnancy. Transitions during pregnancy cycles could also result in alterations to the mammary microenvironment, involving cell autonomous and non-autonomous regulatory cues to sustain MECs response to consecutive pregnancy signals. To dissect the microenvironment's role in post-pregnancy MEC response to pregnancy hormones, we used mammary fat-pad transplantation assays. Cleared fat-pads from pre-pubescent, virgin female mice were transplanted with either pre- or post-pregnancy CD1d+ MaSCs, which have increased mammary reconstitution activity in fat-pad transplants[27]. Recipient female mice (2-month post transplantation) were exposed to pregnancy hormones for 6 days, followed by histological analysis of their mammary glands (Supplementary Fig. 2a). Dissociated and flow cytometer analyzed mammary tissue transplanted with either pre- or post-pregnancy MaSCs showed comparable ratios of luminal and myoepithelial cells after tissue engraftment, suggesting that pregnancy did not affect lineage commitment and differentiation in transplanted MECs (Supplementary Fig. 2b). Histology of transplanted glands from mice during EPH demonstrated that transplantation of post-pregnancy CD1d+ MaSCs yielded a 1.4-fold greater increase in ductal structures ($668 \pm 32$) than glands transplanted with pre-pregnancy CD1d+ MaSCs ($467 \pm 18$), suggesting that post-pregnancy MECs retain their ability to react more robustly to pregnancy signals even after fat-pad transplantation (Fig. 1f, g).

Enhanced branching morphogenesis in response to re-exposure to pregnancy hormones was also recapitulated in in vitro cultures of murine mammary organoids. Post-pregnancy mammary organoids cultured with estrogen, progesterone, and prolactin hormones (complete medium) displayed a 2.3-fold greater number of branching organoids compared with pre-pregnancy cultures (Fig. 1h, i). Furthermore, additional analysis demonstrated increased *Csn2* mRNA levels (10-fold) and increased CSN2 protein levels (approximately 4-fold), in post-pregnancy organoids cultured with pregnancy hormones compared with pre-pregnancy organoids grown under the same hormone conditions (Fig. 1j, Supplementary Fig. 2c–d, and Supplementary Table 1). Given that Csn2 was amongst the genes elevated during second EPH (Fig. 1e), our results support that

cell-autonomous signals control phenotypic and molecular alterations in response to re-exposure pregnancy hormones.

**cMYC overexpression and mammary premalignant lesion development.** Pregnancy decreases mammary tumor frequency in mouse models of mammary oncogenesis[17,20–22]. Several of these studies utilized mammary gland-specific promoters, such as MMTV and WAP-CRE, to drive oncogene expression and tumor development. However, these promoters are enhanced by signals present during pregnancy and lactation[28–30], thus potentially masking epigenomic and transcriptomic changes associated with early oncogenesis and pregnancy-induced protection. To overcome this problem, we utilized a mouse model overexpressing *cMYC* using the CAG promoter, which is independent of pregnancy/lactation signals, under the control of DOX (CAGMYC, Supplementary Fig. 3a).

Nulliparous CAGMYC female mice died after greater than 8 days of DOX treatment, consistent with prolonged *cMYC* overexpression being deleterious to animal health[31]. Thus, to investigate cMYC-driven oncogenesis in live, healthy animals, we analyzed mammary glands from CAGMYC female mice after 2 (DD2) or 5 days (DD5) of DOX treatment. DOX treatment induced substantial histo-pathological alteration to the mammary gland, including flattening of ductal structures and moderate (DD2) to severe, diffuse (DD5) epithelial hyperplasia with atypia, alterations frequently observed in premalignant mammary lesions in mice[32] (Fig. 2a, right panels). None of these alterations were seen in the control CAG-only transgenic mice (Fig. 2a, left panel). Analysis of cytokeratin composition in CAGMYC female mice revealed a progressive expansion of cytokeratin 8 (KRT8) expressing cells, a hallmark of luminal-like cells[33], over the course of the DOX treatment (Fig. 2b). This phenotype was accompanied by the progressive thinning of the basal-like cells (cytokeratin 5, KRT5), often observed during mammary tissue hyperplasia (Fig. 2b).

Given that cMYC-driven mammary tumors may show pathological and transcriptional heterogeneity, we asked which transcriptomic alterations were induced by *cMYC* overexpression in during the establishment of premalignant lesions. Total CAGMYC MECs were isolated from nulliparous mice during sustained *cMYC* overexpression (DD2 and DD5) and analyzed using RNA-seq. In DD2 MECs, we observed enrichment for pathways involving cellular metabolism, such as mitochondrial function and gene splicing, in contrast to pathways upregulated in DD5 MECs, which are associated with control of cell communication processes (Fig. 2c). These results suggest a progressive alteration of transcriptional programs by *cMYC* overexpression, which associates with the initial stages of oncogenesis in

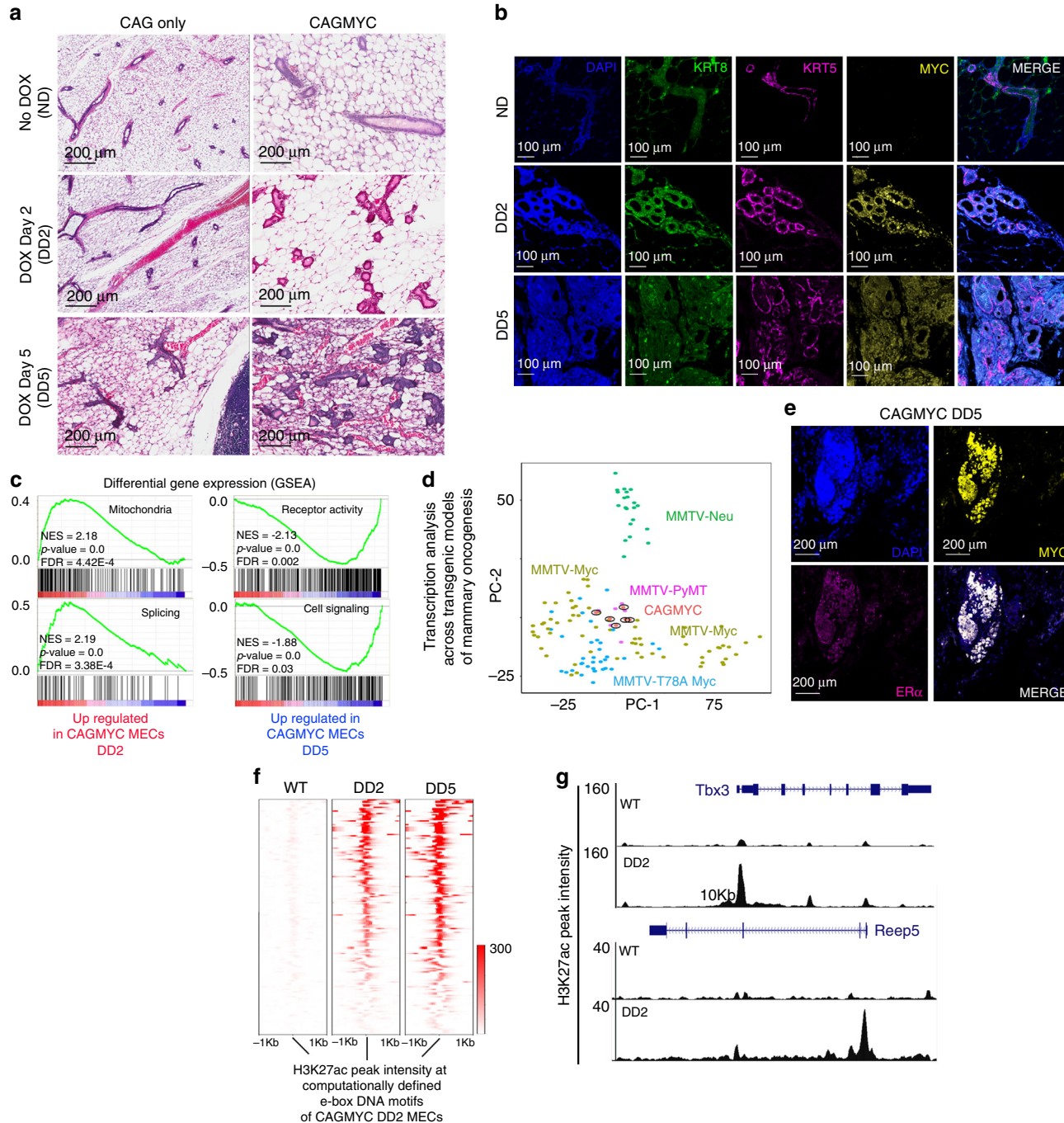

**Fig. 2 cMYC overexpression and mammary premalignant lesion development. a** H&E-stained mammary gland images from nulliparous CAG only and CAGMYC female mice, with no DOX treatment (ND) and treated with DOX for 2 days (DD2) and for 5 days (DD5). Scale: 200 μm. **b** Immunofluorescence images of mammary glands from nulliparous CAGMYC female mice, with no DOX treatment (ND), and treated with DOX for 2 (DD2) and 5 days (DD5), visualizing DAPI (blue), KRT8 (green), KRT5 (magenta), and cMYC (yellow). Scale: 100 μm. **c** GSEA analysis of transcriptional programs enriched in CAGMYC total MECs harvested from nulliparous female mice treated with DOX for 2 days (DD2) and for 5 days (DD5). NES normalized enrichment score. **d** Principal component analysis of gene expression levels from CAGMYC total MECs harvested from nulliparous female mice treated with DOX, compared with publicly available gene expression datasets generated using tumor tissue from transgenic mouse models of mammary tumorigenesis. **e** Immunofluorescence images of mammary glands from nulliparous CAGMYC female mice treated with DOX for 5 days (DD5), visualizing DAPI (blue), ERα (magenta), and cMYC (yellow). Scale: 200 μm. **f** Density plot showing H3K27ac peak intensity at computationally defined e-box DNA binding motifs in MECs harvested from CAGMYC nulliparous female treated with DOX for 2 days (DD2) and 5 days (DD5). **g** Genome browser tracks showing distribution of H3K27ac peaks for Tbx3 and Repp5 genomic loci in WT and DD2 CAGMYC MECs.

mammary glands. We also utilized a transcriptomic approach to classify CAGMYC premalignant lesions according to those from classical models of mammary oncogenesis[34]. We found that the transcriptional profiles of CAGMYC MECs clustered closely with those from luminal-like mammary tumors, including MMTV-PyMT and MMTV-Myc models (Fig. 2d and Supplementary Fig. 3b). Tissue staining with antibodies against estrogen receptor alpha (ERα), a marker for common luminal-like tumor subtypes, demonstrated that CAGMYC MECcomprised malignant lesions with positive ERα receptor nuclear staining, supporting their luminal-like classification (Fig. 2e and Supplementary Fig. 3c).

To investigate the effects of short-term cMYC overexpression on the epigenome of MECs we mapped the active enhancer landscape (H3K27ac ChIP-seq) of total CAGMYC MECs. Many of the H3K27ac peaks present in DD5 CAGMYC MECs (96%) were also present in DD2 CAGMYC MECs, suggesting that development of premalignant mammary lesions largely rely on programs activated during the initial response to cMYC over-expression (Supplementary Fig. 3d, e).

cMYC deletion in mice impaired ductal alveolar genesis during pubescence and pregnancy, indicating its requirement for normal mammary gland development[35]. Thus, we asked whether cMYC overexpression activated a defined set of regulatory regions in MECs undergoing premalignant development. We focused on the gain of H3K27ac at genomic regions recognized by cMYC (e-boxes)[36] in response to cMYC overexpression. Roughly, 4500 H3K27ac peaks were detected at e-boxes in wild-type (WT), non-transgenic MECs and in CAGMYC MECs, suggesting that a set of cis-regulatory elements are activated in MECs independently of cMYC overexpression (Supplementary Fig. 3f). In contrast, we detected a defined set of e-boxes that gained H3K27ac peak intensity in response to cMYC overexpression (Fig. 2f). Genome browser tracks illustrate increased H3K27ac levels in MECs after induction of cMYC overexpression, at cMYC downstream targets Tbx3 and Reep5, both of which have been implicated in mammary oncogenesis[37,38] (Fig. 2g). Thus, short-term cMYC overexpression activates specific epigenomic and transcriptional networks, and causes alterations to tissue morphology resembling those of murine mammary oncogenesis.

**The effects of *cMYC* overexpression on post-pregnancy MECs.** To investigate the effects of cMYC overexpression on post-pregnancy MECs, we treated parous CAGMYC female mice with DOX for 5 days (Supplementary Fig. 4a). Histological analysis revealed that mammary glands of nulliparous female mice displayed a ductal content 3-fold higher than mammary glands from parous CAGMYC female mice, which remained largely unaffected by cMYC overexpression (Fig. 3a, b). In agreement, mammary glands from parous CAGMYC female mice showed tissue morphology and duct numbers (276 ± 42) similar to those from the DOX-treated, CAG-only control group (382 ± 4), supporting that post-pregnancy mammary glands retained a mostly normal phenotype in response to cMYC overexpression (Supplementary Fig. 4b, c). These phenotypic differences were not caused by inefficient transgene induction, as pre- and post-pregnancy CAGMYC MECs expressed comparable cMYC mRNA and protein levels (Fig. 3c and Supplementary Fig. 4d).

To explore whether the resistance to premalignant lesion development was driven by cell-autonomous or non-autonomous mechanisms, we transplanted pre- and post-pregnancy CAGMYC CD1d+ MaSCs into the fat-pads of nulliparous, CAG-only female mice, followed by DOX treatment (Supplementary Fig. 4e). In response to cMYC overexpression, mammary fat-pads transplanted with pre-pregnancy CAGMYC MECs demonstrated severe complex epithelial hyperplasia with atypia and abnormal

ductal morphology, in contrast to fat-pads transplanted with post-pregnancy CAGMYC MECs, which displayed mostly normal tissue histology and lacked abnormal ductal structures (Fig. 3d, e). There were no significant differences on the total number of ducts, or cMYC protein levels, in glands transplanted with either pre-pregnancy (217 ± 49 ducts) or post-pregnancy (140 ± 9 ducts) CAGMYC MECs, suggesting that the lack of abnormal ductal clusters in the post-pregnancy condition was not an artifact associated with transplantation of cMYC overexpressing cells (Fig. 3e, f). Extending cMYC overexpression to 30 days (DD30) also failed to induce the development of premalignant lesions in mammary glands transplanted with post-pregnancy CAGMYC CD1d+ MaSCs, in contrast to glands transplanted with pre-pregnancy CAGMYC CD1d+ MaSCs, which progressed from epithelial hyperplasia to undifferentiated carcinoma lesions (Fig. 3g). Collectively, these results are consistent with cMYC overexpression being less efficient at driving malignant transformation of post-pregnancy MECs.

To investigate whether the cell-autonomous, hyperplasia-reduced phenotype of post-pregnancy CAGMYC MECs would persist under in vitro growth conditions, we utilized mammary organoid cultures. Analysis of pre- and post-pregnancy CAGMYC organoid cultures exposed to increasing concentrations of DOX demonstrated similar induction of cMYC protein levels (Supplementary Fig. 4f, g). Morphological analysis of untreated organoid cultures revealed that pre- and post-pregnancy CAGMYC organoid cultures displayed similar morphology, with pre-pregnancy organoids displaying a higher incidence of normal branching (39 organoids, 35% of total organoids) than post-pregnancy organoids (10 organoids, 8% of total organoids), possibly due to differences on cell culture adaptation (Fig. 3h—left panel, Supplementary Fig. 4h).

DOX treatment of the organoid cultures resulted in abnormal branching of pre-pregnancy CAGMYC organoids, marked by increased cell density in the center of the organoids, a phenotype 7.9-fold (DD1) and 3.1-fold (DD2) reduced in post-pregnancy CAGMYC organoids (Fig. 3h, i). Pre-pregnancy CAGMYC organoids were also 2.9-fold larger than post-pregnancy CAGMYC organoids (Fig. 3j), further supporting that cell-autonomous signals present in post-pregnancy CAGMYC MECs impact the development of premalignant phenotypes in response to cMYC overexpression.

**Post-pregnancy MECs have limited response to *cMYC* overexpression.** To define the transcriptional output of post-pregnancy MECs in response to cMYC overexpression, we carried out unsupervised gene expression analysis on DOX-treated, pre- and post-pregnancy CAGMYC MECs (Supplementary Fig. 5a). cMYC overexpression did not alter lineage-specific transcription, as post-pregnancy luminal and myoepithelial cells clustered together with their pre-pregnancy counterparts (Fig. 4a). Analysis of the parity-associated factors (Supplementary Fig. 1b) showed that 19% and 23% of these gene signatures remain upregulated in post-pregnancy luminal and myoepithelial CAGMYC MECs, respectively, suggesting that pregnancy-associated transcription signatures are not substantially altered by cMYC overexpression (Supplementary Fig. 5b, c).

Unbiased differential gene expression analysis demonstrated downregulation cMYC target genes, and genes associated with responses to estrogen in post-pregnancy CAGMYC MECs (Fig. 4b). Comparing gene expression levels of cMYC-associated genes in CAGMYC luminal MECs with those in WT, non-transgenic luminal MECs, revealed increased levels of mRNA in both pre- and post-pregnancy CAGMYC MECs (Log2Fold-Change >1), demonstrating that cMYC overexpression was

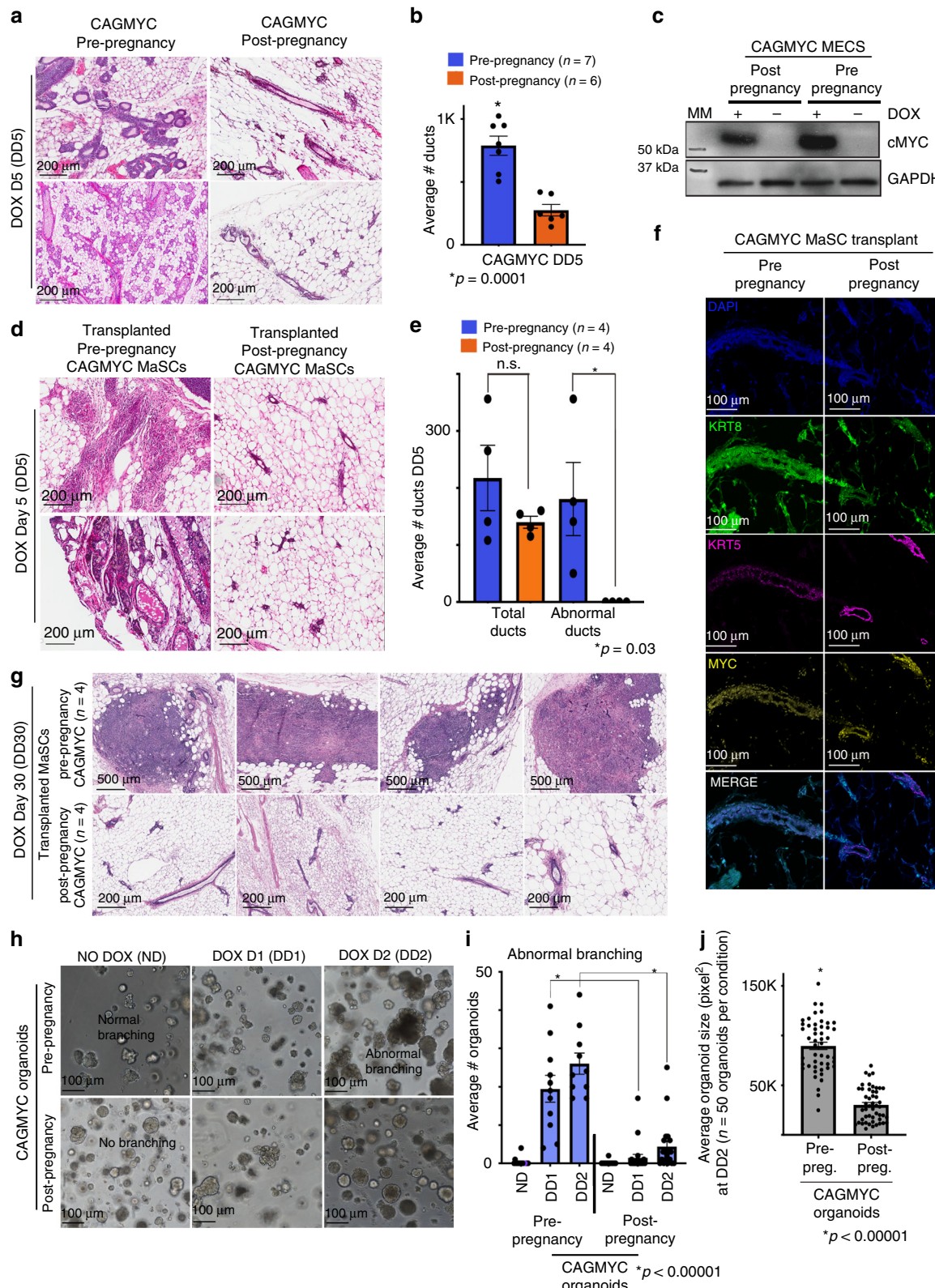

induced successfully in both conditions (Supplementary Fig. 5d, left panel). However, the levels of cMYC-induced gene expression were approximately 2-fold higher in pre-pregnancy CAGMYC MECs, relative to post-pregnancy CAGMYC MECs (Supplementary Fig. 5d), supporting that *cMYC* overexpression is less efficient at activating transcriptional programs in MECs that have been exposed to a full pregnancy cycle.

We next compared the effects of *cMYC* overexpression on the active (H3K27ac) regulatory landscape of total pre- and post-pregnancy CAGMYC MECs. *cMYC* overexpression induced a 6-fold increase in H3K27ac signal intensity at promoter regions in pre-pregnancy CAGMYC MECs, compared with promoter regions from pre-pregnancy WT MECs (Fig. 4c). Conversely, the effect of *cMYC* overexpression on promoter regions was not

**Fig. 3 The effects of *cMYC* overexpression on post-pregnancy MECs. a, b** H&E-stained images and duct quantification from mammary glands harvested from nulliparous (right panel) and parous (left panel) CAGMYC female mice treated with DOX for 5 days (DD5). Scale: 200 μm. $n = 7$ pre-pregnancy mammary glands (left bar) and $n = 6$ post-pregnancy mammary glands (right bar). Bars indicate mean number of ducts. *$p = 0.0001$. **c** Western blot of cMYC protein (62 kDa) in pre-pregnancy and post-pregnancy CAGMYC total MECs, with and without DOX treatment (5 days). GAPDH (146 kDa) used as endogenous control. MM = molecular marker. **d, e** H&E-stained mammary glands images and duct quantification from nulliparous CAG-only control mice, transplanted with pre-pregnancy and/or post-pregnancy CAGMYC CD1d+ MaSCs and treated with DOX for 5 days. Scale: 200 μm. $n = 4$ mammary glands injected with pre-pregnancy CD1d+ MaSCs, and $n = 4$ mammary glands injected with post-pregnancy CD1d+ MaSCs. Bars indicate mean number of total ducts (left) and abnormal ducts (right). n.s. not significant. *$p = 0.003$. **f** Immunofluorescence images of mammary glands from nulliparous CAG-only control mice, transplanted with pre-pregnancy and post-pregnancy CAGMYC MaSCs and treated with DOX for 5 days, visualizing DAPI (blue), KRT8 (green), KRT5 (magenta), and cMYC (yellow). Scale: 100 μm. **g** H&E-stained mammary glands images from nulliparous CAG-only control mice, transplanted with pre-pregnancy and/or post-pregnancy CAGMYC CD1d+ MaSCs and treated with DOX for 30 days. Scale: 500 μm. $n = 4$ mammary glands injected with pre-pregnancy CD1d+ MaSCs, and $n = 4$ mammary glands injected with post-pregnancy CD1d+ MaSCs. **h–j** Representative images, branching quantification, and size quantification of mammary organoid culture of pre- and post-pregnancy CAGMYC MECs, grown with essential media, with or without DOX (0.5 μg/mL). Scale: 200 μm. $n = 2$ independent biological replicates and three technical replicates per experiment. **i** *$p < 0.00001$. **j** $n = 100$ organoids, *$p < 0.00001$. For all analysis, error bars indicate standard error of mean across samples of same experimental group. $p$ values were defined using Student $t$ test.

as strong in post-pregnancy CAGMYC MECs, which displayed 3-fold less H3K27ac signal intensity compared with those of pre-pregnancy CAGMYC MECs (Fig. 4c). This differential response to *cMYC* overexpression was also reflected on the total number of detected H3K27ac peaks, with 26% ($n = 890$) mapping to promoter regions in pre-pregnancy, compared with 6% ($n = 295$) of promoter regions in post-pregnancy CAGMYC MECs (Supplementary Fig. 5e). Conversely, a larger percentage of H3K27ac peaks from post-pregnancy CAGMYC MECs mapped to genic regions (80%), compared with pre-pregnancy CAGMYC MECs (60%), consistent with pregnancy-induced expansion of putative cis-regulatory regions in MECs (Fig. 1), which was not significantly altered by *cMYC* overexpression (Supplementary Fig. 5e). Analysis of H3K27ac intensity levels at PIEs demonstrated retained, parity-induced high H3K27ac levels in post-pregnancy CAGMYC MECs, indicating *cMYC* overexpression did not perturb pregnancy-induced epigenomics signatures (Fig. 4d).

We next asked whether H3K27ac signals would be differentially enriched at e-box DNA motifs in pre- and post-pregnancy CAGMYC MECs. Post-pregnancy CAGMYC MECs displayed weaker H3K27ac peak intensity at e-boxes (~4000 regions) in response to *cMYC* overexpression compared with pre-pregnancy CAGMYC MECs (Supplementary Fig. 5f). In agreement, chromatin accessibility analysis (ATAC-seq) demonstrated a decrease in accessible chromatin at e-box regions in post-pregnancy CAGMYC MECs (Fig. 4e). Additional chromatin accessibility analysis identified a discrete number of enhancer regions exclusive to post-pregnancy CAGMYC MECs ($n = 3248$ ATAC-seq peaks), which were associated with biological processes controlled by pregnancy signals (Supplementary Fig. 5g, h), further supporting that post-pregnancy CAGMYC MECs transcriptome and epigenome are not substantially altered by *cMYC* overexpression.

In order to compare changes in H3K27ac levels with cMYC DNA occupancy we analyzed the Epha2 gene, which codes for a tyrosine receptor kinase expressed in mammary tumors[39], and found it to display decreased H3K27ac intensity in response to *cMYC* overexpression in post-pregnancy CAGMYC MECs (Supplementary Fig. 6a). ChIP-qPCR of Epha2 and Tbx3 genes (Supplementary Fig. 3f) revealed a approximately 3-fold higher cMYC DNA occupancy in pre-pregnancy CAGMYC MECs, further supporting that cMYC is less efficient at associating with chromatin at these genomic regions in post-pregnancy CAGMYC MECs (Supplementary Table 2 and Supplementary Fig. 6b, c).

To analyze the genome-wide distribution of cMYC-chromatin occupancy in pre- and post-pregnancy CAGMYC MECs, we utilized Cleavage Under Targets and Release using Nuclease (Cut&Run). Unbiased transcription factor (TF) DNA-motif analysis revealed enrichment of e-box motifs within the chromatin peaks co-immunoprecipitated with cMYC, suggesting concordance between the observed regions of cMYC-chromatin occupancy (cMYC peaks), and DNA motifs recognized by cMYC (Supplementary Table 3). Differential Cut&Run peak analysis revealed an approximately 3-fold reduction in cMYC peaks in post-pregnancy CAGMYC MECs ($n = 337$ regions), compared with those found in pre-pregnancy CAGMYC MECs ($n = 1127$ regions), supporting that *cMYC* overexpression is less efficient in altering the epigenome of post-pregnancy CAGMYC MECs (Fig. 4f). GO term analysis of cMYC peaks enriched in prepregnancy CAGMYC MECs demonstrated association with pathways that regulate insulin receptor activity, a process regulated by cMYC during cellular malignant transformation[40] (Supplementary Fig. 6d). cMYC peaks enriched in post-pregnancy CAGMYC MECs were associated with genes that promote decreased tumorigenesis and autophagy[41] (Supplementary Fig. 6e, f and Fig. 4g). Expression analysis of genes associated with autophagy and senescence, a byproduct of autophagy processes[42], confirmed their upregulation in post-pregnancy CAGMYC MECs (Supplementary Fig. 6g, h). Specifically, we found increased mRNA levels of the tumor suppressor gene Ecrg4, a factor downregulated in breast cancer tissue[43], in post-pregnancy CAGMYC MECs (~6 Log2FoldChange), and downregulation of Bcl2L12 and Tbx3 mRNAs, factors whose low expression has been correlated with a senescence-like state[44,45].

Further analysis to define whether *cMYC* overexpression results in a senescence-like phenotype, demonstrated that DOX-treated, post-pregnancy CAGMYC organoids express decreased levels of STAT3 protein, downregulation of which induces premature senescence[46], and increased levels of p53 protein, a master regulator of senescence[47] (Supplementary Fig. 6i). We also detected decreased levels of p300 and acetyl p300 proteins specifically in organoids derived from post-pregnancy CAGMYC MECs, a histone acetyltransferase responsible for H3K27ac catalysis, and expressed at low levels in senescent cells[48], indicating that parity may sensitize MECs to a cMYC-induced pre-senescence state (Fig. 4h).

Pregnancy-induced expansion of the H3K27ac landscape controls tissue development in response to pregnancy signals, and could also play a role in regulating autophagy and senescence, therefore interfering with *cMYC* overexpression-driven phenotypes associated with malignant transformation of MECs. Moreover, it was recently shown that knockdown of residual p300 levels in senescent cells suppressed the expression

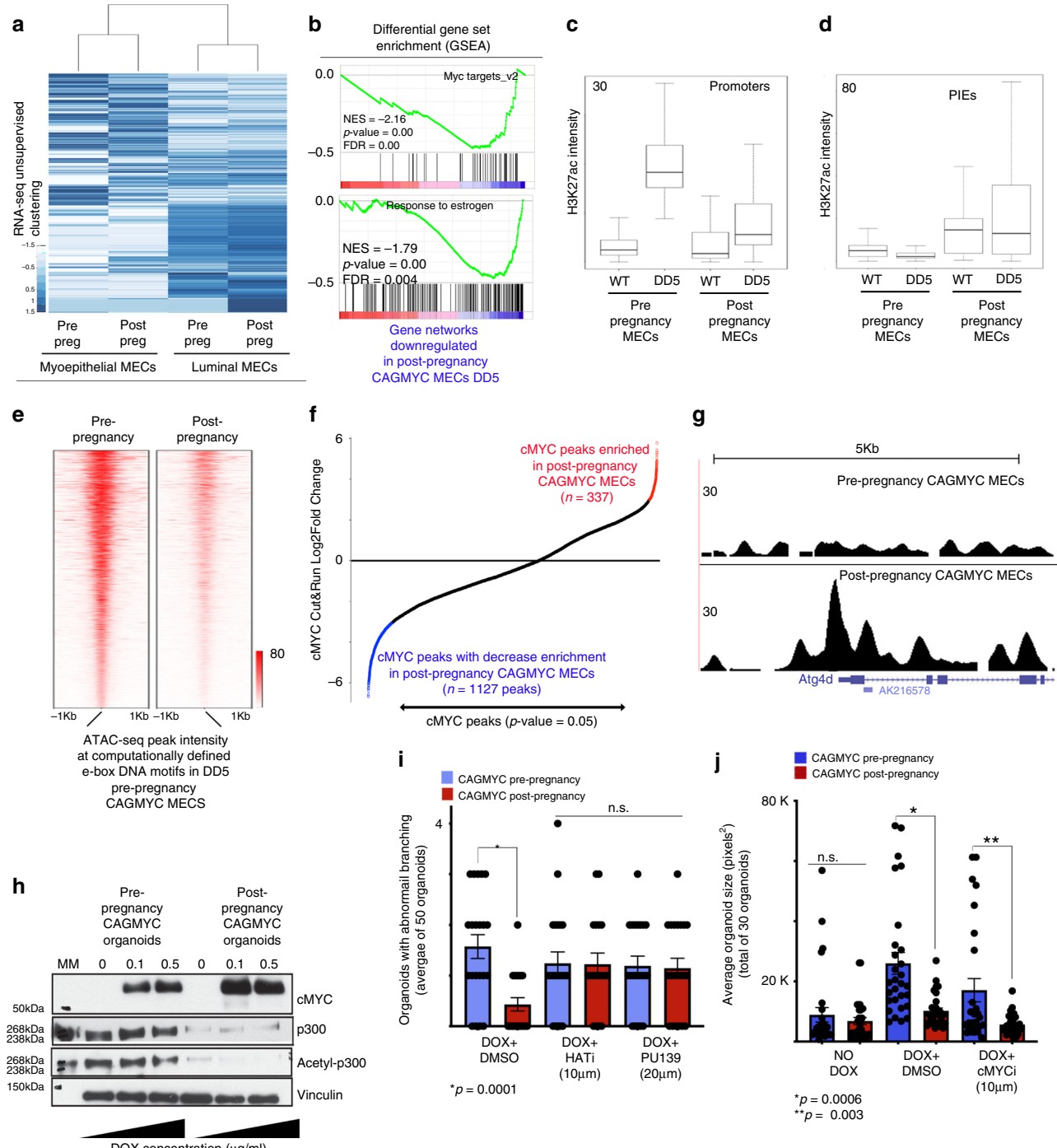

of senescence-related genes, thus reverting their senescent state[49]. To test the idea that post-pregnancy CAGMYC MECs can autonomously resist perturbation by cMYC-driven oncogenic programs in a p300/senescence-dependent manner, we treated CAGMYC organoid cultures with small-molecule inhibitors that block p300-histone acetyltransferase. Treatment of organoids with the inhibitors PU139 and HATi resulted in a 2.6-fold and 2.4-fold increase in abnormal branching in post-pregnancy CAGMYC organoid cultures, suggesting that inhibition of p300 increased adverse phenotypic changes in response to *cMYC* overexpression (Fig. 4i and Supplementary Fig. 6j). These alterations were not dependent on blocking cMYC activity, given that pre-pregnancy CAGMYC organoids remained 2.9-fold larger than organoids derived from CAGMYC post-pregnancy MECs,

after treatment with cMYC inhibitor (cMYCi) (Fig. 4j and Supplementary Fig. 6k), thus supporting that specific perturbations to post-pregnancy CAGMYC MECs can revert their ability to respond to *cMYC* overexpression, and engage on malignant transformation.

## Discussion

Our results revealed epigenetic alterations to the MEC regulatory landscape that enable reactivation of pregnancy-induced programs in response to pregnancy hormones. These programs influenced the development of premalignant lesions in response to oncogene overexpression. Our epigenetic and transcriptomic analyses corroborate histological and cellular data, in both

**Fig. 4 Post-pregnancy MECs have limited response to *cMYC* overexpression. a** Gene expression hierarchical clustering of DOX-treated (DD5), FACS-isolated, pre- and post-pregnancy CAGMYC MECs. **b** GSEA analysis of gene networks downregulated in DOX-treated, FACS-isolated, pre- and post-pregnancy CAGMYC MECs. NES normalized enrichment score. **c, d** Averaged H3K27ac intensity at (**c**) promoter regions or (**d**) parity-induced elements (PIEs) of FACS-isolated, pre- and post-pregnancy WT and CAGMYC MECs (DD5). Error bars represent the variation of H3K27ac intensity at analyzed regions. Center line represents the median of the dataset. Bounds of the box represent the 25th (lower bound) and 75th percentile (upper bound). Whiskers represent the minimum and maximum of the non-outlier data. **e** Density plot showing computationally defined e-box DNA binding motifs with high ATAC-seq peak intensity in pre-pregnancy CAGMYC (DD5), compared with ATAC-seq peak intensity at same e-box DNA binding site in post-pregnancy CAGMYC MECs (DD5). **f** cMYC Cut&Run peak enrichment analysis, showing peaks enriched (red) or depleted (blue) in post-pregnancy CAGMYC MECs (DD5). $p$ value = 0.05 or lower. **g** Genome browser tracks showing cMYC occupancy in DOX-treated, pre- and post-pregnancy CAGMYC MECs. **h** Western blot of cMYC, p300, and acetyl-p300 proteins in organoid cultures derived from pre-pregnancy and post-pregnancy CAGMYC MECs, with and without DOX treatment (2 days). Vinculin protein levels were used as endogenous control. MM = molecular marker. **i** Number of branched organoids from pre- and post-pregnancy CAGMYC MECs organoid cultures, grown with essential media and DOX (2 days, 0.5 µg/mL), with and without histone acetyltransferase inhibitors (HATi = HAT Inhibitor II 10 µM and PU139 20 µM). $n = 30$ organoids, *$p = 0.0001$; n.s. = no statistically significant differences. **j** Size quantification of pre- and post-pregnancy CAGMYC mammary organoids, grown with essential media and DOX (5 days, 0.5 mg/mL), with and without cMYC inhibitor (cMYCi = cMyc inhibitor (10058-F4) 10 µM). $n = 30$ organoids, *$p = 0.0006$; **$p = 0.002$. For all analysis, error bars indicate standard error of mean across samples of same experimental group. $p$ values were defined using Student $t$ test.

mammary gland tissue and organoid systems. These findings demonstrate that cell autonomous signals present in post-pregnancy MECs regulate gene expression, cellular activation, and resistance to malignant transformation.

Previous studies of rodent and human post-pregnancy MECs revealed parity-modulated factors and signaling networks that may contribute to breast cancer risk[11,50,51]. Here, we have used a similar approach, but have instead focused on the active, regulatory, epigenetic landscape of pre- and post-pregnancy murine MECs, with the goal of understanding its role in transcriptional activation, not only in response to signals from consecutive pregnancies but also during the initial stages of oncogenesis. Considering the possibility that not all malignant lesions may progress into fully developed tumors, our findings suggest that systems that suppress cancer development may be engaged in post-pregnancy MECs, and block cancer initiation.

Interestingly, the expansion of the pregnancy-induced enhancer landscape minimally recapitulated the transcriptional output of post-pregnancy MECs during tissue homeostasis (non-pregnancy state), as demonstrated by our gene expression analyses (Fig. 1). Thus, the chromatin state we assayed, the H3K27ac activation mark, may not be sufficient to discriminate between enhancer regions that were once highly active (during pregnancy), and those that acquired a less active/resting state after parity, poised to respond to future pregnancy signals. It is also possible that the abundance of specific TFs and the activity of epigenetic factors fluctuate across non-pregnancy, pregnancy, and post-pregnancy states, guiding additional chromatin remodeling and gene expression control. Given it is less clear how post-pregnancy cells reorganize their epigenome, we have focused our analyses on the dynamics of enhancer activation and gene regulation in response to pregnancy and the early stages of oncogenesis, revealing the complexity of the networks involved in these events.

Our epigenomic and transcriptomic analyses utilized a cell isolation strategy previously applied to isolate-defined MEC populations[12,27]. However, we cannot exclude that, after pregnancy, these cell-surface markers recognize a more diverse cell population compared with those existing prior to pregnancy. Previous studies have reported parity-induced MECs[5,52,53], and recent single-cell RNA-seq analyses demonstrated alterations to MEC populations throughout gestation, lactation, and involution stages of mammary gland development[54]. However, DNA methylation analyses of several mammary cell types from fully involuted, post-pregnancy mammary glands have demonstrated that alterations to the epigenome were, to some extent, shared by

most mammary cell types[12], suggesting that pregnancy-induced epigenomic alterations may not be restricted to MEC lineage identities.

Our present study also revealed the relationship between *cMYC* overexpression, pregnancy-induced epigenomic alterations, and tissue/cellular abnormalities. cMYC, a potent oncogene, is overexpressed in ~50% of human breast cancers[55]. In addition, several oncogenic signaling networks (Brca1 loss, MAPK-Ras hyperactivation, PI3K-AKT/PKB hyperactivation) converge on the oncogenic potential of deregulated *cMYC* expression[56]. Furthermore, chromosomal abnormalities found in *cMYC* overexpressing mouse models are syntenically and developmentally comparable to those of human breast cancer[57]. Therefore, an inducible *cMYC* overexpression model system to understand mammary malignant initiation may help identify molecular targets for validation in human breast specimens, improving understanding of mammary oncogenesis.

It is important to note that signals driving early oncogenesis may differ from those present after disease establishment. Our analysis of pre-pregnancy CAGMYC MECs suggests that transcriptional programs and the epigenome are differentially regulated across the *cMYC* overexpression timeline, supporting the notion that longer exposure to *cMYC* overexpression may reprogram breast epithelial cellular identity[58]. Nonetheless, we have shown that cMYC-driven signals do not fully induce epigenomic and transcriptomic alterations that support malignant transformation of post-pregnancy mammary glands. This is also seen in transplantation assays, where *cMYC* overexpressing post-pregnancy CAGMYC MECs did not progress into malignant transformation.

But how can pregnancy decrease mammary oncogenesis? Previous studies established that both a full pregnancy cycle, and an induced pseudo-pregnancy decreased the frequency of mammary tumors in several mouse strains, including in chemically induced mammary tumorigenesis models[17,21], and those accompanied by MMTV-driven *cMYC* overexpression[22]. Conversely, lack of active p53 is associated with the development of mammary tumors in murine[20,59] and human association studies[60], suggesting that a decrease in p53 dosage in post-pregnancy MECs may promote cancer initiation. Our study revealed a substantial increase of p53 protein in post-pregnancy CAGMYC organoids, possibly promoting a senescent state that could block the development of malignant phenotypes.

It is also possible that pregnancy may induce alterations that influence the mammary gland stroma and/or its immune composition. Pregnancy-induced alterations to the mammary gland

ECM have been suggested to play a role in preventing the progression of established cancer cells[61–63]. Alterations to immune composition during post-pregnancy mammary gland involution have also been suggested to influence mammary tumor progression[64]. Interactions among the stroma, immune system and the MEC epigenome should be explored to define their roles in blocking the early onset of oncogenesis in a post-pregnancy setting, and to reveal their intersection with the parity-induced epigenomic changes we describe.

Importantly, our mammary organoid experiments confirmed the cell-autonomous characteristics of pregnancy-induced changes, and their ability to block responses to *cMYC* overexpression. Using this system, we confirmed post-pregnancy MECs are less responsive to *cMYC* overexpression. Our analysis also demonstrated the utility of the organoid system for perturbing signals present in post-pregnancy MECs. Utilizing combinations of small-molecule inhibitors and genetic manipulations in this system will enable the identification of signals that either promote or block responses to oncogenic signals in a target-specific manner.

Finally, our data revealed the stability of the pregnancy-induced epigenome, both in in vivo and in vitro pregnancy-naïve environments, emphasizing the cell-autonomous nature of the altered post-pregnancy epigenome. Such in vivo and in vitro strategies will be required to monitor the dynamics of enhancer activity and transcriptional regulation in a pregnancy-hormone dependent fashion. Ultimately, the use of in vitro organoid strategies may allow for better understanding of enhancer activation, transcriptional regulation, and responses to oncogenes in breast tissue obtained from women with various reproductive histories, with the goal to identify and characterize the human-specific features of pregnancy-induced developmental dynamics in cancer predisposition.

## Methods

**Mouse lines**. Balb/C and C57/BL6 female mice were purchased from Charles River. CAGs-rtTA3 mice (B6N.FVB(Cg)-Tg(CAG-rtTA3)4288Slowe/J, the Jackson Laboratory) and tetO-MYC mice (FVB/N-Tg(tetO-MYC)36aBop/J, the Jackson Laboratory) were crossed for the establishment of CAGMYC transgenic mouse strain (129/C57BL6 background). All animals were housed at a 12 light/12 dark cycle, with a controlled temperature of 72 °F and 40–60% of humidity. All experiments were performed in agreement with approved CSHL Institutional Animal Care and Use Committee.

**Mammary gland isolation**. Mammary glands were harvested and processed as fully described in "Supplementary Methods." In short, mammary glands were harvested and digested into a single-cell suspension. MECs were separated from immune cells (CD45+), red blood cells (TER119+), endothelial cells (CD31+), and fibroblasts (CD34+) using lineage depletion antibodies and MACS magnetic column (Miltenyi Biotech). Lineage-depleted (LIN−) MECs were utilized as described in "Supplementary Methods."

**Illumina library preparation and NextGen sequencing**. FACS-isolated MECs were utilized for the preparation of NextGen Illumina libraries, as described in "Supplementary Methods."

**Histological analysis**. Tissue histology and immunofluorescence staining (IF) were performed as described in "Supplementary Methods."

**Reporting summary**. Further information on research design is available in the Reporting Summary linked to this article.

## Data availability

All data publicly accessible through NCBI databases [https://www.ncbi.nlm.nih.gov/]. RNA-seq, ChIP-seq, ATAC-seq, and Cut&Run datasets are available in the BioProject database under number PRJNA544746 [https://www.ncbi.nlm.nih.gov/bioproject/PRJNA544746]. Results described on Fig. 1a, b, prepregnancy MECs RNA-seq are found in the BioProject database under number PRJNA192515 [https://www.ncbi.nlm.nih.gov/bioproject/?term=PRJNA192515]. Results described on Fig. 2d, a total of 157 mammary tumor tissue samples can be retrieved within the GEO Profiles database under numbers GSE13221 [https://www.ncbi.nlm.nih.gov/geo/query/acc.cgi?acc=GSE13221], GSE15904

[https://www.ncbi.nlm.nih.gov/geo/query/acc.cgi?acc=GSE15904], and GSE30805 [https://www.ncbi.nlm.nih.gov/geo/query/acc.cgi?acc=GSE30805].

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

## Acknowledgements

This work was performed with assistance from the CSHL Flow Cytometry Shared Resources, CSHL Animal Facility, the CSHL NextGen Sequencing Shared Resources, and the CSHL Histology Shared Resource, which are supported by the CSHL Cancer Center Support Grant 5P30CA045508. This work was financially supported by the Manhasset Women's Coalition Against Breast Cancer, the Glen Cove Cares Foundation, the CSHL and Northwell health affiliation, the Rita Allen Scholar Award, the V-foundation Scholar Award, the AACR-Breast Cancer Research Foundation Award, the Pershing Square Sohn Prize for Cancer Research, and the NIH/NCI grant R01CA248158-01 (C.O.d.S.). This work was also supported with editing services provided by Dr Riddihough from Life Science Editors.

## Author contributions

C.O.d.S. designed and supervised the research; C.O.d.S, M.A.M., and M.J.F. wrote the manuscript. C.O.d.S., M.A.M., and M.C.T. performed bioinformatics analysis; C.O.d.S., M.J.F., C.C., S.L.C., S.-T.Y., M.F.C., and W.D.F. performed experiments and analyzed results; J.E.W. supported histology analysis and provided pathological assessment of all images.

## Competing interests

The authors declare no competing interests.
