## [Peer Review File · Nature Communications]

Reviewers' comments:

Reviewer #1 (Remarks to the Author):

This paper presents data that a first pregnancy in mice induces a long-lasting reorganization of the mouse mammary epithelial cells (MEC) epigenome (an "epigenetic memory"). Post-pregnancy epigenomic modifications, in particular H3K27ac peaks, were 8 fold higher and found preferentially at enhancer regions that drive specific genes networks in response to a second pregnancy. The epigenetic data correlated with gene expression as genes associated with Parity Induced Elements (PIEs) showed increased expression by RNA-seq in MEC from mice in the 2nd pregnancy cycle, indicative of an "epigenetic memory". Additionally, the authors report that a 5 day exposure to cMYC overexpression in transgenic nulliparous mice led to invasive malignant tumors, but had no such effect on parous mice. Rather, the post-pregnancy epigenome was incompatible with the epigenomic reorganization driven by cMYC overexpression, and blocked the activation of MYC-downstream signals that drive proliferation networks of mammary epithelial cells. The possibility that alterations to the post-pregnancy microenvironment might be responsible for the observed changes was examined by fat-pad transplantation assays, using a different mouse strain as transplant recipient (Balb/C) than the prior experiments (FVB). The results supported their hypothesis that intrinsic, cell-autonomous signals influence post-pregnancy MECs, including resistance to cMyc-induced malignancy. Studies in 3D organoid culture also supported their "epigenetic memory" hypothesis. The authors hypothesize that pregnancy-induced changes alter the ability of cMYC overexpression to drive malignant transformation of MECs, and may be a significant consequence of remodeling dynamics that prevent cancer development.

This review is mainly concerned with the use of the terms "model" and "model system" (presumably reference to the murine systems employed modeling human) in this report. The authors' assumption is that page (p)4 "our study represents the initial step towards understanding the predisposition to breast cancer, in a pregnancy-dependent fashion." However, there is no discussion of how well mice do or do not model humans in the processes examined, and significant data exist to question this assumption. While the data on epigenetic memory associated with murine pregnancy is of interest, and suggests approaches for studying human MEC, the data on cMYC induction of malignancy is not in concert with what is known about human breast carcinogenesis.

In a revision, the authors should be clear that the results presented are relevant to mice, and avoid suggestions that these data are a "model" of the human situation, or a "model system", unless they have some relevant data in women. Since reduction mammoplasty tissues from women of all ages and parity are available, it is possible to obtain some basic information in humans, rather than just leveraging this work to study other "relevant mouse models of mammary tumorigenesis". It is known that the process of malignant progression is significantly different between large long-lived organisms such as humans and small short-lived organisms such as mice. Extensive studies from the Seluanov and Gorbunova groups (e.g., NatRevCancer 2018) have shown that mice do not fully repress telomerase activity even in adult cells, and can readily transform to immortality (e.g., 3T3) and malignancy. In contrast, it has been suggested that immortalization is a rate-limiting step in human carcinogenesis, and spontaneous transformation to even immortality is virtually unknown in normal human adult cells. Published studies (Garbe CellCycle 2014) indicate that overexpression of cMyc does not cause cultured normal human MEC to become immortal or malignant, but cMyc can immortalize, without malignancy, already aberrant finite human MEC. In general, MYC has been associated with induction of telomerase activity in humans – a process with radically different behavior in human vs. mouse adult cells. Altogether, these data do not support the mouse being an accurate "model system" of human for the studies being reported, and all such implications that a "model" is being presented should be removed. e.g., p4 "we set out to characterize the epigenetic remodeling dynamics in mammary epithelial cells and their link to murine cancer initiation and prevention." "we established that pregnancy assigns cell autonomous features on the epigenome of murine mammary epithelial cells,"; "our

findings suggest that pregnancy-induced mammary cancer prevention relies on the epigenomic changes in MECs"; p7 "These findings suggest that short-term cMYC overexpression is sufficient to drive oncogenesis of murine mammary epithelial cells."; p10 "If the pregnancy-induced mammary cancer protection phenotype is restricted to post-pregnancy MECs"; p14 "Collectively, these results highlight a complex network supported by cMYC overexpression at initial stages of murine mammary oncogenesis and demonstrate how such programs are not activated in post-pregnancy MECs."; p15 "decreased the frequency of mammary tumors in several murine systems including those lacking p53 expression [24], models of with chemically-induced mammary tumorigenesis [15, 17] and models accompanied by with cMYC overexpression [54]. In our inducible murine system ..."; p15 "Most importantly, our studies revealed a series of pathways and factors that are specifically repressed in post-pregnancy CAGMYC MECs, which can be validated and harnessed as the basis for molecularly designed strategies to block murine mammary tumor development."

The reduction in mammary cancer due to a full first term pregnancy is modest in both mice and humans (from the authors p3: "a full-term pregnancy early in a woman's life before the age of 25 can reduce the risk of breast cancer by one third"). Even this data is not so straight-forward, as early first pregnancy can also initially lead to an increased risk of breast cancer (Borges & Schedin, Cancer 2012). As such, one would not expect to see a dramatic molecular change, such as the results reported here for cMYC in mice, to be responsible for these observations in women.

Additionally, to what extent do the molecular changes of pregnancy in mice reflect what is known in humans? Such information would be valuable for evaluating the relevance of this work to humans.

The authors should also be clearer in the results section what mouse strains are being used, what is known about their susceptibility to mammary cancer and any significant differences between the 2 strains used, and briefly what cells types are being isolated and examined.

p4: "Most importantly, our study represents the initial step towards understanding the predisposition to breast cancer, in a pregnancy-dependent fashion.". How is that so – there is no connection made to the molecular processes in humans, or any comparison of how mice and humans differ in both cancer progression and in hormonal expression/morphological organization during development, estrous, and pregnancy.

p5: the author could be clearer why the increased H3K27ac "within genic and intergenic regions" "is likely to represent distal enhancer elements"

p6: "we generated RNA-seq libraries from sorted MECs isolated from female mice." Sorted for what/how?

p8: "Consistent with the phenomenon that pregnancy elicits a protective effect against breast cancer, we anticipated that cMYC oncogenesis would be thwarted in a post-pregnancy environment." Why this anticipation? The majority of human breast cancers (presumably many from parous women since most are in older women) show either cMyc amplification or overexpression. Other than the changes associated with hTERT reactivation and immortalization, increased cMyc expression is among or the most common change in human breast cancer cells compared to normal – data at odds with what is reported here for murine mammary cancer and cMyc induced carcinogenesis.

It is also questionable how treating an adult transgenic mouse with a 5 day exposure to increased cMyc relates to the role of a persistent change in cMyc expression in promoting human breast cancer. cMyc has been published to play a role in conferring immortality to aberrant finite human MEC – but murine cells do not have a comparable barrier to immortality for cMyc to act on.

Thus, this paper should strictly confine conclusions to murine MEC.

Reviewer #2 (Remarks to the Author):

The authors present a manuscript that tackles an interesting question, how are the long lasting effects of pregnancy protective for breast cancer? They have hypothesized that this protective effect is due to epigenetic changes in the genome. To test this hypothesis, they generated a new tet responsive bigenic strain of mice expressing the human form of cMyc. They showed that there was a resistance to transformation in the model system with induction of lactation and that there were key differences in epigenetic remodeling in the post-pregnant state. The finding that Myc was occupying the EphA2 locus in pre but NOT in post pregnancy was a very nice validation of the global work at a single gene scale. Together the manuscript provides an interesting perspective on a complex problem.

Major points;

1 – Implied throughout the manuscript is the idea that the CAGMYC mice are oncogenic – going so far as to compare the organoids to MMTV-PyMT organoids. However, the manuscript lacks a description of the actual tumors that arise in this new strain of mice. Do these mice form tumors? There are many GEM mice modeling Myc expression from the original MMTV-Myc from the Leder lab, to the Chodosh inducible Myc mice, to knock-ins at the Rosa26 locus from the Sears lab – how does this strain compare for tumor latency and histological subtypes?

1A – Continuing in this thought, the work here showed that Myc overexpression failed to drive the transformation of post-pregnant mammary epithelial cells. The ultimate test though, would be to determine if these mice that undergone lactation cycles were blocked from mammary tumorigenesis.

1B – Still thinking about tumor formation, much data exists for various MMTV transgenics showing that tumors developed earlier in the cohorts that were allowed to progress through multiple lactation / involution cycles. While this has traditionally been thought to be primarily due to the large increase in glucocorticoid regulated transcriptional induction of the transgene during pregnancy and lactation, the argument made in this manuscript was that mammary epithelial cells failed to respond to Myc overexpression. How do the authors reconcile their findings with the prior data showing that MMTV-Myc transgenics that had undergone lactation develop tumors more quickly than the virgin counterparts?

2 – Work from Pepper Schedin's group has established the initial increased risk of breast cancer associated with pregnancy with a crossover to a protective effect after several years. This should be discussed in relation to the findings here.

Minor points:

1 – The generation of the model and some of the characterization should be moved out of the supplemental figures and into the main body – specifically Supplemental 2A, 2F and an example from 2C. This could go into Figure 2.

Reviewer #3 (Remarks to the Author):

This manuscript reports important epigenomic modifications in the mouse mammary gland that are involved in protection of the postparturiant gland from risk of developing mammary tumors. cMYC plays a critical role in this risk. Although I found this area of investigation to be important and novel, it was also without detail throughout. This work would be difficult to repeat in another lab, as written.

Major issues to be addressed:

1. Study design was not mentioned. Was there no consideration for age matching the mice involved (collection at same ages for accurate comparisons)? Was estrous stage considered in mice at collection? It is not clear if the appropriate control or even strain of mouse was used in all experiments, how tissues from same mice were used or if the tissues can be compared across endpoints. Also, there is no mention of the number of animals used in each of these studies, the take-rate (fail-rate) of transplants, how many animals were evaluated in transplant studies, and what endpoints were used a priori to estimate the n needed. I am assuming that the reason there are no statistics shown for the differences in physiology that you report is that you couldn't perform them due to low power. I note that n is mentioned in two of the subsections of the supplemental data, but is a major deficit of the paper elsewhere.

2. The figures do not stand on their own. There is not enough detail in the figures to know what is shown (with a couple of exceptions in the supplement). For example - Fig 1. what is the difference between post pregnancy and 1st pregnancy and 2nd pregnancy in ages and at what stage of these 'life stages' were the animals necropsied? Are they comparable? In Figure 2, what are the IDs of the samples shown vertically? Also, in Supp Fig 2C, the IDs horizontally. It says they are representative images - of what, if numerous slides of same are shown. Was the histology that different amongst mice of same group? If so, even more reason to state the n in all studies. In the heat maps, there is no information on the grouping of genes that were different between cMYC or life stages.

3. There is little if any detail on the findings in this paper in the abstract and results section. You show data for specific genes in your figures, yet never mention them in the text - there is no quantification of anything, just differences noted. The intro and discussion really don't say much that isn't already said in the very wordy results section. In fact, it would be helpful if the preamble to the CAGMYC mice was in the introduction, as it appears that the mice were bought from Jackson Labs (even though you may have developed them) - you make it sound like they were developed within this study, and there is no methods section to back that up. The results section could be compressed to half its size, then insert the details of what you found. Also, why you jumped right to H3K27ac and not something else (or a different global analysis) is not clear and this should show up in the introduction. There is no mention of why the mouse strains you chose are relevant to humans (some more is needed in third paragraph of intro). But, also, 3 of the 7 short intro paragraphs are conclusions.

4. There are no study limitations mentioned. Most all who study the mammary gland know the importance of considering the epithelium and stromal portions of the gland. It is fine that the authors focused on the epithelium for this project, but it should be clear what if any role the stromal/fat fraction may play (especially as it pertains to transplants) and that your findings are likely epithelia specific - an interesting topic in the discussion might be what role the stroma/fat may play in modulating this effect as that fraction of the gland is also heavily modified following a pregnancy.

Also, in the results you seem to conclude that the mammary gland of the FVB strain may act differently toward pseudo-pregnancy than other strains (may call this a strain-specific effect and potentially list other strains you compared to). If that is the case, why were transplants made into cleared fat pads of Balb/C mice and pseudo-pregnancy induced in that strain??

I hope that many of these issues might be addressed as this is an important area of study.

February 21st 2019

RE: NATURE COMMUNICATION - NCOMMS-18-30590.

"Pregnancy reprograms the enhancer landscape of mammary epithelial cells and alters the response to cMYC driven oncogenesis"

First, we would like to thank the editorial board and reviewers for their valuable comments (blue font). We have included our responses below (black font) to each of the points and indicated the location of each modification in the original manuscript. We have fully revised the manuscript for clarity and informational accuracy. We have also included additional analyses addressing many of the reviewer's requests. Overall, we appreciate the editor's continued consideration of our manuscript.

Reviewers' comments:

Reviewer #1

This paper presents data that a first pregnancy in mice induces a long-lasting reorganization of the mouse mammary epithelial cells (MEC) epigenome (an "epigenetic memory"). Post-pregnancy epigenomic modifications, in particular H3K27ac peaks, were 8-fold higher and found preferentially at enhancer regions that drive specific genes networks in response to a second pregnancy. The epigenetic data correlated with gene expression as genes associated with Parity Induced Elements (PIEs) showed increased expression by RNA-seq in MEC from mice in the 2nd pregnancy cycle, indicative of an "epigenetic memory". Additionally, the authors report that a 5-day exposure to cMYC overexpression in transgenic nulliparous mice led to invasive malignant tumors, but had no such effect on parous mice. Rather, the post-pregnancy epigenome was incompatible with the epigenomic reorganization driven by cMYC overexpression, and blocked the activation of MYC-downstream signals that drive proliferation networks of mammary epithelial cells. The possibility that alterations to the post-pregnancy microenvironment might be responsible for the observed changes was examined by fat-pad transplantation assays, using a different mouse strain as transplant recipient (Balb/C) than the prior experiments (FVB). The results supported their hypothesis that intrinsic, cell-autonomous signals influence post-pregnancy MECs, including resistance to cMYC-induced malignancy. Studies in 3D organoid culture also supported their "epigenetic memory" hypothesis. The authors hypothesize that pregnancy-induced changes alter the ability of cMYC overexpression to drive malignant transformation of MECs, and may be a significant consequence of remodeling dynamics that prevent cancer development. This review is mainly concerned with the use of the terms "model" and "model system" (presumably reference to the murine systems employed modeling human) in this report. The authors' assumption is that page (p)4 "our study represents the initial step towards understanding the predisposition to breast cancer, in a pregnancy-dependent fashion. "However, there is no discussion of how well mice do or do not model humans in the processes examined, and significant data exist to question this assumption. While the data on epigenetic memory associated with murine pregnancy is of interest, and suggests approaches for studying human MEC, the data on cMYC induction of malignancy is not in concert with what is known about human breast carcinogenesis.

Point #1: In a revision, the authors should be clear that the results presented are relevant to mice, and avoid suggestions that these data are a “model” of the human situation, or a “model system”, unless they have some relevant data in women. Since reduction mammoplasty tissues from women of all ages and parity are available, it is possible to obtain some basic information in humans, rather than just leveraging this work to study other “relevant mouse models of mammary tumorigenesis”.

Answer to Point #1: We thank the reviewer for the comment. Although, traditionally mice have been utilized as an appropriate system for modeling human disease, we acknowledge that the term “model system” can be seen as an over-representation of our results. Therefore, and in order to address this major point, we made the following modifications to the manuscript:

a) Terminology in reference to mouse studies: We have corrected instances where we refer to our studies as “model” or “model system” throughout the manuscript. We now clearly state that our findings can only explain pregnancy blocking initial malignancy in the utilized transgenic mouse strain. We have also removed references that indicate that the transgenic mouse strain utilized fully resembles human disease, a point that we agree can be misleading.

b) Relevance of findings in human tissue: We agree with the reviewer that the next immediate step is to translate our findings to human tissue. Given that pregnancy substantially alters the enhancer landscape of mouse mammary epithelial cells, and that such regulatory regions lay on partially syntenic genomic regions, one would have to employ more population evolution analysis to define the commonalities between the species. We have recently engaged in collaborations with evolutionary geneticists to the evolutionary aspects of pregnancy-induced epigenetic reprogramming.

Point #2: It is known that the process of malignant progression is significantly different between large long-lived organisms such as humans and small short-lived organisms such as mice. Extensive studies from the **Seluanov** and **Gorbunova** groups (e.g., NatRevCancer 2018) have shown that mice do not fully repress telomerase activity even in adult cells, and can readily transform to immortality (e.g., 3T3) and malignancy. In contrast, it has been suggested that immortalization is a rate-limiting step in human carcinogenesis, and spontaneous transformation to even immortality is virtually unknown in normal human adult cells. Published studies (Garbe CellCycle 2014) indicate that overexpression of cMYC does not cause cultured normal human MEC to become immortal or malignant, but cMYC can immortalize, without malignancy, already aberrant finite human MEC. In general, MYC has been associated with induction of telomerase activity in humans – a process with radically different behavior in human vs. mouse adult cells. Altogether, these data do not support the mouse being an accurate “model system” of human for the studies being reported, and all such implications that a “model” is being presented should be removed. e.g., p4 “we set out to characterize the epigenetic remodeling dynamics in mammary epithelial cells and their link to murine cancer initiation and prevention.” “we established that pregnancy assigns cell autonomous features on the epigenome of murine mammary epithelial cells,”; “our findings suggest that pregnancy-induced mammary cancer prevention relies on the epigenomic changes in MECs”; p7 “These findings suggest that short-term cMYC overexpression is sufficient to drive oncogenesis of murine mammary epithelial

cells.”; p10 “If the pregnancy-induced mammary cancer protection phenotype is restricted to post-pregnancy MECs”; p14 “Collectively, these results highlight a complex network supported by cMYC overexpression at initial stages of murine mammary oncogenesis and demonstrate how such programs are not activated in post-pregnancy MECs.”; p15 “decreased the frequency of mammary tumors in several murine systems including those lacking p53 expression [24], models of with chemically-induced mammary tumorigenesis [15, 17] and models accompanied by with cMYC overexpression [54]. In our inducible murine system ...”; p15 “Most importantly, our studies revealed a series of pathways and factors that are specifically repressed in post-pregnancy CAGMYC MECs, which can be validated and harnessed as the basis for molecularly designed strategies to block murine mammary tumor development.”

Answer to Point #2: We thank the reviewer for the perceptive comment. We agree with the reviewer that several statements made throughout the manuscript might result in misleading interpretations, implying stronger parallels to human breast biology than was intended. Thus, we have carefully clarified the specific points raised by the reviewer, in addition to any others that we judged necessary to clarify and simplify the description of our results.

Point #3: The reduction in mammary cancer due to a full first term pregnancy is modest in both mice and humans (from the authors p3: “a full-term pregnancy early in a woman’s life before the age of 25 can reduce the risk of breast cancer by one third”). Even this data is not so straightforward, as early first pregnancy can also initially lead to an increased risk of breast cancer (Borges & Schedin, Cancer 2012). As such, one would not expect to see a dramatic molecular change, such as the results reported here for cMYC in mice, to be responsible for these observations in women. Additionally, to what extent do the molecular changes of pregnancy in mice reflect what is known in humans? Such information would be valuable for evaluating the relevance of this work to humans.

Answer to Point #3: Pregnancy has a dual effect on breast cancer risk. More specifically, all pregnancies, independent of age, result in a short-term increased breast cancer risk (also referred as parity-induced breast cancer) which varies from 5-10 years. However, after the approximately 10-year window following pregnancy, younger mothers (<25 years old) benefit from a decreased risk of developing breast cancer compared to older mothers (>30 years old) and nulliparous females. This is in agreement with Borges and Schedin (Cancer 2012), as they quoted:

*“For first-time mothers aged 25 years or younger, **the risk is modestly increased** compared to nulliparous women, and in a large Norwegian cohort, has been shown to **last approximate 9 years**, at which time a cross-over effect occurs. **This cross-over effect then changes the role of pregnancy from one of breast cancer promotion to subsequent protection.** For a woman who delays childbearing until age 30 to 35, **the risk for breast cancer is significantly increased compared to younger mothers**, and the cross-over effect is delayed until her 60s. Women who wait until age >35 years for their first childbirth permanently increase their risk of breast cancer compared to nulliparous women. **Peak incidence of breast cancer does not occur during pregnancy or in the immediate postpartum period, but rather approximately 6 years postpartum.**”*

Several studies, utilizing microarrays (Russo et al, 2008, 2012; Blakely, et al. 2006) and promoter-focused DNA methylation analysis (Choudhury, et al. 2013) have observed substantial differences in gene expression and the epigenome among nulliparous and parous mice, rats and humans. However, a complete explanation for their relevance to decreasing breast cancer risk was not addressed. In addition, none of these datasets are publicly available, further restricting our ability to re-analyzed and compare to our data.

Nonetheless, given the importance for clarification of all these points raised by the reviewer, we extended the discussion of such points in the current manuscript (Lines: 54-62).

Point #4. The authors should also be clearer in the results section what mouse strains are being used, what is known about their susceptibility to mammary cancer and any significant differences between the 2 strains used, and briefly what cells types are being isolated and examined.

Answer to Point #4: We thank the reviewer for bringing our attention to the missing information in the Result section. We have revised the entire manuscript for clarity and informational accuracy, and have included substantial experimental details for each of the experiments presented.

Point #5: “Most importantly, our study represents the initial step towards understanding the predisposition to breast cancer, in a pregnancy-dependent fashion.” How is that so – there is no connection made to the molecular processes in humans, or any comparison of how mice and humans differ in both cancer progression and in hormonal expression/morphological organization during development, estrous, and pregnancy.

Answer to Point #5: We thank the reviewer for giving us the chance to clarify this statement. The events described in our study are in relation to epigenetic changes modulated by murine pregnancy. We have clarified our statements throughout the manuscript to better reflect our data and avoid unintentional implications.

Point #6: the author could be clearer why the increased H3K27ac “within genic and intergenic regions” “is likely to represent distal enhancer elements”

Answer to Point #6: We thank the reviewer giving us the chance to clarify the statement. We have revised the manuscript to better describe the rationale of our study and clarify the description of our findings. (Lines: 98-101; 113-135).

Point #7: “we generated RNA-seq libraries from sorted MECs isolated from female mice.” Sorted for what/how?

Answer to Point #7: We utilized classical cell surface markers of luminal and myoepithelial mammary cells for FACS sorting. We have included a graphic to Sup. Figures 1, 3 and 4 to illustrate the strategy for cell isolation and analysis, in addition to including an extended description of the methodology in the Methods and Supplementary Methods sections.

Point #8: “Consistent with the phenomenon that pregnancy elicits a protective effect against breast cancer, we anticipated that cMYC oncogenesis would be thwarted in a post-pregnancy environment.” Why this anticipation? The majority of human breast cancers (presumably many

from parous women since most are in older women) show either cMYC amplification or overexpression. Other than the changes associated with hTERT reactivation and immortalization, increased cMYC expression is among or the most common change in human breast cancer cells compared to normal – data at odds with what is reported here for murine mammary cancer and cMYC induced carcinogenesis.

Answer to Point #8: We thank the reviewer giving us the chance to clarify the statement. The assumption that “*cMYC oncogenesis would be thwarted in a post-pregnancy environment*”, as we quoted on our original submission, was based on the notion that pregnancy decreases tumor frequency in transgenic mice. However, none of these studies investigated the effects of cMYC on altering the epigenome. Our study aimed to understand how the enhancer landscape of post-pregnancy MECs respond to cMYC over-expression during early onset of oncogenesis. Unfortunately, we currently cannot provide explanations for the potential divergent effects of cMYC over-expression during early stages of oncogenesis in murine or human MECs. We have revised the entire manuscript to clarify our hypothesis and conclusions to better represent our findings.

Point #9: It is also questionable how treating an adult transgenic mouse with a 5 day exposure to increased cMYC relates to the role of a persistent change in cMYC expression in promoting human breast cancer. cMYC has been published to play a role in conferring immortality to aberrant finite human MEC – but murine cells do not have a comparable barrier to immortality for cMYC to act on.

Answer to Point #8: We thank the reviewer for pointing this out. Our studies focused on the initial molecular changes driven by cMYC over-expression that precede oncogenesis, a poorly-studied topic. This approach allowed us to gain an understanding of how influential the post-pregnancy epigenome is on the mammary gland. Our studies did not focus on, or rule out, additional molecular alterations brought on by a sustained cMYC overexpression that could contribute to oncogenesis. In fact, results in Sup.Fig.2 demonstrate tissue alterations and gene expression changes following cMYC overexpression in nulliparous female mice. We have included text in the main manuscript to better explain our study and clarify our rationale.

Reviewer #2

The authors present a manuscript that tackles an interesting question, how are the long lasting effects of pregnancy protective for breast cancer? They have hypothesized that this protective effect is due to epigenetic changes in the genome. To test this hypothesis, they generated a new tet responsive bigenic strain of mice expressing the human form of cMYC. They showed that there was a resistance to transformation in the model system with induction of lactation and that there were key differences in epigenetic remodeling in the post-pregnant state. The finding that Myc was occupying the EphA2 locus in pre but NOT in post pregnancy was a very nice validation of the global work at a single gene scale. Together the manuscript provides an interesting perspective on a complex problem.

Major points:

Point #1 – Implied throughout the manuscript is the idea that the CAGMYC mice are oncogenic – going so far as to compare the organoids to MMTV-PyMT organoids. However, the manuscript lacks a description of the actual tumors that arise in this new strain of mice. Do these mice form tumors? There are many GEM mice modeling Myc expression from the original MMTV-Myc from the Leder lab, to the Chodosh inducible Myc mice, to knock-ins at the Rosa26 locus from the Sears lab – how does this strain compare for tumor latency and histological subtypes?

Answer to Point #1: We thank the reviewer for the comment and for encouraging us to better describe the transgenic mouse utilized in our studies. As mentioned by the reviewer, the effects of cMYC on mammary tumor development, in pregnancy-dependent and independent studies, have been described by several other groups. In these studies, MMTV- and WAP-CRE promoters were utilized to drive oncogene expression. However, MMTV-promoter activity (used by the Leder lab and Chodosh lab) and WAP-CRE activity (used by the Sears lab) are enhanced by signals present during pregnancy and lactation. Therefore, we judged these transgenic lines less suitable to study the effects of pregnancy-induced epigenome on gene regulation, and its associated with mammary tumor predisposition.

Nonetheless, and in order to address the questions regarding mammary tumorigenesis, we included the following additional data in the manuscript:

Transcriptional analysis across transgenic mouse strains programmed to undergo mammary oncogenesis. We took a transcriptomic approach to compare DOX-treated CAGMYC transgenic mice and the more “classical” transgenic strains programmed to undergo mammary oncogenesis. In doing so, we collected publicly available, global gene-expression datasets from MMTV-PyMT, MMTV-Neu, and two variations of MMTV-cMYC transgenic mammary tumors (Fig. 2d and Sup. Fig. 2c). Our analysis demonstrated that DOX-treated CAGMYC transgenic MECs carry a distinct expression profile from those harvested from MMTV-Neu, a transgenic strain that develops mammary tumors resembling human breast cancer subtypes over-expressing HER2. Conversely, DOX-treated CAGMYC transgenic MECs shared a transcriptional output similar to those displayed by MMTV-PyMT MECs (luminal-like mammary tumor) and MMTV-cMYC MECs (mixed mammary tumors, with MECs spanning luminal-like and basal-like characteristics). These results were in agreement with the immunofluorescence data presented in Fig.2b, which shows DOX-treated, CAGMYC mammary tissue expressing both luminal markers (KRT8) and basal markers (KRT5). Altogether, we conclude that the early onset of oncogenesis observed in DOX-treated CAGMYC transgenic mammary glands likely resembles a mixed mammary tumor subtype, with both luminal-like and basal-like characteristics (Lines: 207-211).

Estrogen Receptor α (ER α) staining. To confirm the potential mixed phenotype of DOX-treated CAGMYC mammary tissue, we utilized canonical Estrogen Receptor α (ER α) staining that defines luminal-like mammary tumor subtypes (Sup. Fig.2d). In agreement with our transcriptomic analysis, we found that ER α staining had a scattered distribution across DOX-

treated CAGMYC mammary tissue, further suggesting the presence of both luminal-like cells (ER α positive) and ER α negative cells. (Lines: 212-217).

Point #2 – Continuing in this thought, the work here showed that Myc overexpression failed to drive the transformation of post-pregnant mammary epithelial cells. The ultimate test though, would be to determine if these mice that undergone lactation cycles were blocked from mammary tumorigenesis.

Answer to Point #2: Our work did not evaluate tumor development in parous transgenic mice. The main goal of our work was to define the effects of cMYC overexpression that occur during the early onset of mammary oncogenesis, its effect on mammary epigenome, and the epigenetic changes that are associated with blocking early cancer progression. It is known that pregnancy decreases the frequency of mammary tumors in other transgenic mouse strains. Our work further reveals the breast cancer preventive effects of pregnancy, in addition to tackling a unique set of issues related to epigenomic changes and unresponsiveness to oncogenesis.

Point #3 – Still thinking about tumor formation, much data exists for various MMTV transgenics showing that tumors developed earlier in the cohorts that were allowed to progress through multiple lactation / involution cycles. While this has traditionally been through to be primarily due to the large increase in glucocorticoid regulated transcriptional induction of the transgene during pregnancy and lactation, the argument made in this manuscript was that mammary epithelial cells failed to respond to Myc overexpression. How do the authors reconcile their findings with the prior data showing that MMTV-Myc transgenics that had undergone lactation develop tumors more quickly than the virgin counterparts?

Answer to Point #3: We thank the reviewer for the comment. As mentioned above (answer to point 1) the mammary gland-specific promoters MMTV and WAP, used to understand the role of cMYC on murine mammary oncogenesis, both display baseline activity during pubescent development of mammary glands, and this activity is enhanced by signals that peak during pregnancy, such as, hydrocortisone, prolactin, insulin, and estrogen (Wagner et al 2001; Wen et al 1995; Webster et al 1995). Thus, utilization of an inducible system that does not rely on pregnancy-enhanced transgene activation was essential to study the effects of pregnancy on mammary early onset of oncogenesis. In light of the reviewer's comment, we have added text clarifying our justification of using the CAGMYC transgenic mouse (Lines: 71-74; 169-177).

Point #4 – Work from Pepper Schedin's group has established the initial increased risk of breast cancer associated with pregnancy with a crossover to a protective effect after several years. This should be discussed in relation to the findings here.

Answer to Point #4: We thank the reviewer for highlighting the importance of describing both effects of pregnancy on cancer risk. We have included a more in-depth discussion about the subject in the manuscript (Lines: 54-62).

Minor points:

Point #5 – The generation of the model and some of the characterization should be moved out of the supplemental figures and into the main body – specifically Supplemental 2A, 2F and an example from 2C. This could go into Figure 2.

Answer to Point #5: In response to the reviewer's comments, we have restructured all the figures to better improve the message of our manuscript.

Reviewer #3 (Remarks to the Author):

This manuscript reports important epigenomic modifications in the mouse mammary gland that are involved in protection of the postparturiant gland from risk of developing mammary tumors. cMYC plays a critical role in this risk. Although I found this area of investigation to be important and novel, it was also without detail throughout. This work would be difficult to repeat in another lab, as written.

Major issues to be addressed:

Point #1 - Study design was not mentioned. Was there no consideration for age matching the mice involved (collection at same ages for accurate comparisons)? Was estrous stage considered in mice at collection? It is not clear if the appropriate control or even strain of mouse was used in all experiments, how tissues from same mice were used or if the tissues can be compared across endpoints. Also, there is no mention of the number of animals used in each of these studies, the take-rate (fail-rate) of transplants, how many animals were evaluated in transplant studies, and what endpoints were used a priori to estimate the n needed. **I am assuming that the reason there are no statistics shown for the differences in physiology that you report is that you couldn't perform them due to low power.** I note that n is mentioned in two of the subsections of the supplemental data, but is a major deficit of the paper elsewhere.

Answer to Point #1: We thank the reviewer for the comment and agree that a more in-depth description of experimental design and mice number should be included. We fully revised the description of results, figure legends and material and methods section to include the needed information to better appreciate our findings.

Point #2 - The figures do not stand on their own. There is not enough detail in the figures to know what is shown (with a couple of exceptions in the supplement). For example - Fig 1. what is the difference between post pregnancy and 1st pregnancy and 2nd pregnancy in ages and at what stage of these 'life stages' were the animals necropsied? Are they comparable? In Figure 2, what are the IDs of the samples shown vertically? Also, in Supp Fig 2C, the IDs horizontally. It says they are representative images - of what, if numerous slides of same are shown. Was the histology that different amongst mice of same group? If so, even more reason to state the n in all studies. In the heat maps, there is no information on the grouping of genes that were different between cMYC or life stages.

Answer to Point #2: We thank the reviewer for the comment and agree more information is needed to appreciate the data in the figure legends. We revised all figures and figure legends, including the addition of cartoons to better illustrate the experimental details.

Point #3 - There is little if any detail on the findings in this paper in the abstract and results section. You show data for specific genes in your figures, yet never mention them in the text - there is no quantification of anything, just differences noted. The intro and discussion really don't say much that isn't already said in the very wordy results section. In fact, it would be helpful if the preamble to the CAGMYC mice was in the introduction, as it appears that the mice were bought from Jackson Labs (even though you may have developed them) - you make it sound like they were developed within this study, and there is no methods section to back that up. The results section could be compressed to half its size, then insert the details of what you found. Also, why you jumped right to H3K27ac and not something else (or a different global analysis) is not clear and this should show up in the introduction. There is no mention of why the mouse strains you chose are relevant to humans (some more is needed in third paragraph of intro). But, also, 3 of the 7 short intro paragraphs are conclusions.

Answer to Point #3: We thank the reviewer for the helpful comments. We have fully revised the abstract, introduction, results and discussion sections of the manuscript to better clarify our hypothesis and the goals of this study.

Point #4 - There are no study limitations mentioned. Most all who study the mammary gland know the importance of considering the epithelium and stromal portions of the gland. It is fine that the authors focused on the epithelium for this project, but it should be clear what if any role the stromal/fat fraction may play (especially as it pertains to transplants) and that your findings are likely epithelia specific - an interesting topic in the discussion might be what role the stroma/fat may play in modulating this effect as that fraction of the gland is also heavily modified following a pregnancy.

Answer to Point #4: We thank the reviewer for the comment. We have included a discussion of a possible role for stroma/fat influencing our findings (Fig. 4e, Lines: 441-454).

Point #5 - Also, in the results you seem to conclude that the mammary gland of the FVB strain may act differently toward pseudo-pregnancy than other strains (may call this a strain-specific effect and potentially list other strains you compared to). If that is the case, why were transplants made into cleared fat pads of Balb/C mice and pseudo-pregnancy induced in that strain??

Answer to Point #5: We thank the reviewer for bringing up this important point. We have clarified all mouse strains and control animals utilized through the manuscript. (Lines: 182-184; 274-279).

Point #6 - I hope that many of these issues might be addressed as this is an important area of study.

Answer to Point #6: We thank the reviewer for acknowledging the importance of our findings and for the opportunity to revise our manuscript for publication in Nature Communications.

Reviewers' comments:

Reviewer #1 (Remarks to the Author):

This revised paper has greatly improved in response to previous comments about the relevance of these murine studies to human health conditions. A few more minor points remain that should be changed, because this reviewer considers it very important for our understanding of human breast cancer progression that there be no implication that murine mammary cancer progression is an accurate model of the human process. As has been well documented, small short-lived animals lack stringent repression of telomerase in adult cells, and therefore lack the crucial replicative senescence tumor suppressor barrier in progression. As this barrier has been suggested to be rate-limiting in human oncogenesis in vivo and in culture, presenting mice as "models" without qualification has led to a general lack of attention and understanding of the immortalization process that overcomes replicative senescence during human carcinoma progression - a potentially valuable therapeutic target that cannot be studied in mice. Consequently, the following changes should be made:

1) Mouse mammary glands and the cancers derived from them should in all cases be referred to as mammary glands and mammary cancer - not breasts or breast cancer, i.e.

28 pregnancy-induced [breast] mammary cancer prevention involves the epigenomic changes in MECs brought about by pregnancy.

78 Overall, our findings suggest that pregnancy-induced [breast] mammary cancer prevention involves the epigenomic changes in MECs

167 Pregnancy decreases the frequency of mammary tumors in a variety of transgenic [mouse] models of murine mammary oncogenesis [breast cancer] [13-~15], raising the possibility that pregnancy-induced epigenomic and transcriptomic modifications may underlie pregnancy-induced [breast] mammary cancer protection

264 In prior studies, pregnancy-induced [breast] mammary cancer protection was studied in rodents

2) If the authors want to compare mice and humans on the effects of pregnancy on future mammary cancer risk, they should be as specific in the following sentence of exactly what is seen in the mouse, and how that compares to the human time-frame; "similar" is not sufficient, and in any case is only referring to the very long-term reduction, not the immediate long-term (5-10 years) increase in risk. Are there any murine strains that are similar to humans in showing a mouse-equivalent of 5-10 yrs increased mammary cancer incidence after pregnancy? Do the murine strains used show the same modest effect as in humans for very long-term effects ("a full-term pregnancy early in a woman's life before the age of 25 can reduce the risk of breast cancer by one third")? Otherwise, this reference to human should be omitted.

While research has shown increased

57 breast cancer risk for roughly five-ten years after parturition [16-18], there is a long-term reduction of risk of breast cancer for women completing a full-term pregnancy before the age of 30 [13, 14, 18, 19]. A similar risk decrease following pregnancy has been observed in mice, where completion of a pregnancy cycle dampens the frequency of mammary tumor development [14, 20-22].

This paper presents interesting data on the effects of pregnancy in murine systems on epigenetic status and future mammary cancer vulnerability. The relevance to human situations is unknown, particularly in light of MYC overexpression/amplification being present in a majority of human breast cancers, where it can be assumed that a large percentage of these women were parous. Possibly, this is because, unlike in mice, in human mammary epithelial cells MYC has been

implicated in reactivating telomerase and overcoming replicative senescence – necessary for human breast cancer progression.

Reviewer #2 (Remarks to the Author):

I appreciate the care the authors took in responding and revising the manuscript. No other concerns present.

Reviewer #3 (Remarks to the Author):

The authors have nicely addressed most of the comments from the reviewers, especially as it pertained to moving text and word-smithing. However, there remains a problem with the way these studies were powered and analyzed. These are major issues that may not be easily fixed. The authors stated that details on animal numbers were added to the figure legends so that they would stand alone and that is the case for Figure 3 and the supplemental data only. The issues of the quality of the data remain. For instance, in a well powered study, there would be a comparison of the data derived from the studies noted in Suppl table 1 and Suppl figure 1J across cell types (associations at least), but without a scale to indicate gene fold-change and no analyses, these data are not convincing. There are numerous examples of this within the manuscript. Further, some instances state that $n > 5$ were used and there is no evidence that the litter was used as a covariate in these statistical analyses - really a requirement in these types of studies. The design shows no more than 5 animals per group, so $n > 5$ would require advanced analyses. Further, there is no stats section that I could find in this manuscript. Finally, the lack of tumor data or more evidence that the lesions in the glands already observed would form tumors is needed. Many lesions do not go to tumors and may recede.

The observational findings here are interesting and potentially compelling, but there is not strong data presented.

Point-by-point response to the Reviewer's comments

RE: NATURE COMMUNICATION - NCOMMS-18-30590B-Z.

We thank the reviewers for their feedback as well as their suggestions and critiques on our manuscript entitled "Pregnancy reprograms the epigenome of mammary epithelial cells and blocks the development of premalignant lesions in response to cMYC-overexpression". We value this input and have used their remarks to improve our manuscript. We believe we have sufficiently addressed their suggestions and concerns. Below, please find detailed point-by-point responses (in black) to all of the specific comments from the Reviewers (in blue). All of the textual alterations are highlighted in red in both the manuscript and supplementary information files.

Reviewer #1 (Remarks to the Author):

This revised paper has greatly improved in response to previous comments about the relevance of these murine studies to human health conditions. A few more minor points remain that should be changed, because this reviewer considers it very important for our understanding of human breast cancer progression that there be no implication that murine mammary cancer progression is an accurate model of the human process. As has been well documented, small short-lived animals lack stringent repression of telomerase in adult cells, and therefore lack the crucial replicative senescence tumor suppressor barrier in progression. As this barrier has been suggested to be rate-limiting in human oncogenesis in vivo and in culture, presenting mice as "models" without qualification has led to a general lack of attention and understanding of the immortalization process that overcomes replicative senescence during human carcinoma progression - a potentially valuable therapeutic target that cannot be studied in mice. Consequently, the following changes should be made:

Point #1) Mouse mammary glands and the cancers derived from them should in all cases be referred to as mammary glands and mammary cancer – not breasts or breast cancer, i.e. 28 pregnancy-induced [breast] mammary cancer prevention involves the epigenomic changes in MECs brought about by pregnancy. 78 Overall, our findings suggest that pregnancy-induced [breast] mammary cancer prevention involves the epigenomic changes in MECs. 167 Pregnancy decreases the frequency of mammary tumors in a variety of transgenic [mouse] models of murine mammary oncogenesis [breast cancer] [13-15], raising the possibility that pregnancy-induced epigenomic and transcriptomic modifications may underlie pregnancy-induced [breast] mammary cancer protection²⁶⁴ In prior studies, pregnancy-induced [breast] mammary cancer protection was studied in rodents

Response to Point #1) To address this issue, we have now replaced the term "breast" with "mammary" when referring to the mouse model throughout the manuscript.

Point #2) If the authors want to compare mice and humans on the effects of pregnancy on future mammary cancer risk, they should be as specific in the following sentence of exactly what is seen in the mouse, and how that compares to the human time-frame; "similar" is not sufficient, and in any case is only referring to the very long-term reduction, not the immediate long-term (5-10 years) increase in risk. Are there any murine strains that are similar to humans in showing a mouse-equivalent of 5-10 yrs increased mammary cancer incidence after pregnancy? Do the murine strains used show the same modest effect as in humans for very long-term effects ("a full-term pregnancy early in a woman's life before the age of 25 can reduce the risk of breast cancer by one third")? Otherwise, this reference to human should be omitted. While research has shown increased 57 breast cancer risk for roughly five-ten years after

parturition [16-18], there is a long-term reduction of risk of breast cancer for women completing a full-term pregnancy before the age of 30 [13, 14, 18, 19]. A similar risk decrease following pregnancy has been observed in mice, where completion of a pregnancy cycle dampens the frequency of mammary tumor development [14, 20-22].

Response to Point #2) We agree that important differences exist between mouse and human with respect to the kinetics of pregnancy-induced breast cancer protection. We have modified the discussion section of the manuscript to discuss this point.

In regards to the investigation of the short-term effect of pregnancy on increasing mammary tumorigenesis: a series of reports described that MMTV-promoter based transgenic mouse models have increased mammary tumorigenesis after pregnancy, thus representing a transgenic mouse model for the study of pregnancy-associated breast cancer (Kordon, 2008). That perhaps has to do with the fact that MMTV-promoter activity is augmented by signals present during pregnancy and lactation. This is also the main reason for why we decided not to utilize MMTV-promoter based transgenic mouse models for our research.

Point #3) This paper presents interesting data on the effects of pregnancy in murine systems on epigenetic status and future mammary cancer vulnerability. The relevance to human situations is unknown, particularly in light of MYC overexpression/amplification being present in a majority of human breast cancers, where it can be assumed that a large percentage of these women were parous. Possibly, this is because, unlike in mice, in human mammary epithelial cells MYC has been implicated in reactivating telomerase and overcoming replicative senescence – necessary for human breast cancer progression.

Response to Point #3) We agree with the Reviewer that we should be conservative in our interpretation of how our observations in the mouse model will relate to breast cancer progression in human patients. As pointed out by the Reviewer, telomerase function is a critical aspect of MYC function in human breast cancer which is less relevant in our mouse model, owing to the unusually long telomeres in mice. In addition, it is clear that many parous women will develop MYC-driven breast cancer, and hence it should not be interpreted that pregnancy provides a complete blockade of breast cancer initiation. To further clarify these important issues, we have now carefully revised the entire text of our manuscript such that we are not over-interpreting the human relevance of our findings. In addition, we have added a new paragraph to the Discussion section of our manuscript that explicitly addresses the important differences between mouse and human breast/mammary carcinogenesis. Nevertheless, using the experimentally tractable mouse model our study provides fundamental insight into how pregnancy and reprogram the output of cancer-promoting oncogenes in mammary epithelial cells, which we believe provides an important conceptual advance for the field that will guide future experimentation in human breast cancer models.

Reviewer #2 (Remarks to the Author):

Point #1) I appreciate the care the authors took in responding and revising the manuscript. No other concerns present.

Answer to Point #1) We are grateful to the Reviewer for recognizing our efforts to improve our revised manuscript.

Reviewer #3 (Remarks to the Author):

The authors have nicely addressed most of the comments from the reviewers, especially as it pertained to moving text and word-smithing. However, there remains a problem with the way these studies were powered and analyzed. These are major issues that may not be easily fixed.

We have made significant efforts over the past several months to ensure that our study has properly quantified all of our experimental results. These efforts are reflected in all figures and supplemental figures.

Point #1) The authors stated that details on animal numbers were added to the figure legends so that they would stand alone and that is the case for Figure 3 and the supplemental data only.

Response to Point #1) We apologize for the confusion in our earlier revision, in which we did not provide a clear description of animal numbers utilized in all experimental approaches presented in this manuscript. We have now carefully enumerated and corrected every figure legend.

Point #2) The issues of the quality of the data remain. For instance, in a well powered study, there would be a comparison of the data derived from the studies noted in Suppl table 1 and Suppl figure 1J across cell types (associations at least), but without a scale to indicate gene fold-change and no analyses, these data are not convincing. There are numerous examples of this within the manuscript.

Response to Point #2) We have modified the supplementary figure to reflect this concern and have also corrected this specific issue in all the figures.

Point #3) Further, some instances state that $n > 5$ were used and there is no evidence that the litter was used as a covariate in these statistical analyses - really a requirement in these types of studies. The design shows no more than 5 animals per group, so $n > 5$ would require advanced analyses.

Response to Point #3) We have included biological replicates and advanced/supporting analysis to all figures, as described:

Fig.1 - The effects of re-exposure to pregnancy hormone on in vivo and in vitro MEC development. The observation that post-pregnancy MECs respond more robustly to re-exposure to pregnancy hormones was drawn from mammary gland transplantation analysis ($n=3$ mammary glands injected with pre-pregnancy MaSCs, and $n=3$ mammary glands injected with post-pregnancy MaSCs), and supported with advanced analysis utilizing organoid cultures branching analysis ($n=3$ independent biological replicates, with approximately 120 organoids per quantification/replicate), organoid cultures IF analysis, organoid cultures qPCR analysis ($n=3$ independent biological replicates), and organoid cultures flow cytometry analysis ($n=3$ independent biological replicates). Each biological replicate corresponds to mammary tissue from 2 mice (8-10 mammary gland pairs) per task.

Fig.2 - The effects of cMYC overexpression on epigenomic and transcriptomic alterations during mammary premalignant development. The observation that progressive, short-term, cMYC overexpression induced substantial molecular alterations in MECs was drawn from the analysis of RNAseq libraries and H3K27ac ChIP-seq libraries, which were analyzed utilizing $n=2$ biological replicates per task. Each biological replicate corresponds to mammary tissue from 1 mouse (4-5 mammary gland pairs) per task.

Fig.3 - The effects of cMYC overexpression on post-pregnancy MECs premalignant development. The observation that *cMYC* overexpression does not induce the development of premalignant lesions in post-pregnancy mammary glands, was drawn from mammary gland tissue histology analysis (n=7 mammary glands injected with pre-pregnancy MaSCs, and n=6 mammary glands injected with post-pregnancy MaSCs), and supported with advanced analysis utilizing mammary transplantation analysis following DOX treatment for 5 days (n=4 mammary glands injected with pre-pregnancy MaSCs, and n=4 mammary glands injected with post-pregnancy MaSCs), mammary transplantation analysis following DOX treatment for 30 days (n=4 mammary glands injected with pre-pregnancy MaSCs, and n=4 mammary glands injected with post-pregnancy MaSCs), organoid cultures branching analysis (n=3 independent biological replicates, with approximately 100-300 organoids per quantification/replicate), and organoid cultures size analysis (n=3 independent biological replicates, with approximately 50 organoids per quantification/replicate). Each transplantation and organoid biological replicate correspond to mammary tissue from 2 mice (8-10 mammary gland pairs) per task.

Fig.4 - The effects of cMYC overexpression on post-pregnancy MECs premalignant development. The observation that CAGMYC post-pregnancy MECs do not fully respond to epigenetic and transcriptomic alterations brought by *cMYC* overexpression was drawn from the analysis of analysis of RNAseq libraries, H3K27ac ChIP-seq libraries, ATAC-seq libraries, *cMYC* Cut&Run libraries, and organoid cultures, which were analyzed utilizing n=2-3 biological replicates per task. Each biological replicate corresponds to mammary tissue from 1 mouse (4-5 mammary gland pairs) per task.

Point #4) Further, there is no stats section that I could find in this manuscript.

Response to Point #4) This information has been added to the Supplementary Methods.

Point #5) Finally, the lack of tumor data or more evidence that the lesions in the glands already observed would form tumors is needed. Many lesions do not go to tumors and may recede.

Response to Point #5) We have provided data supporting that prolonged *cMYC* overexpression (30 days) results in sustained development of malignant lesions in mammary glands injected with pre-pregnancy MaSCs, in marked contrast to mammary glands injected with post-pregnancy CAGMYC MaSCs which do not display malignant lesions (Fig.3).

Point #6) The observational findings here are interesting and potentially compelling, but there is not strong data presented.

Response to Point #6) We thank the reviewer for finding our research interesting. As previously mentioned, we improved every aspect of this manuscript to improve robustness and to clarify results across all complementary approaches. In addition, we have included 27 additional figure panels and 1 table, which taken together support our main conclusion, as follows:

Fig.1i – Quantification of branching morphogenesis in mammary organoid cultures, demonstrating increased branching in organoids derived from post-pregnancy MECs after pregnancy hormone treatment. n = 3 independent biological replicates. Each biological replicate was generated utilizing mammary organoids pulled from 2 pre-pregnancy female mice and 2 post-pregnancy female mice.

Fig.1j – IF analysis from mammary organoids cultures derived from pre- or post-pregnancy MECs after pregnancy hormone treatment, demonstrating increased CSN2 levels in post-pregnancy mammary organoids.

Sup.Fig.1b – Gene expression analysis, demonstrating previously described parous signature in pre- and post-pregnancy luminal MECs. n=2 RNA-seq biological replicates. Each biological replicate was generated utilizing mammary luminal epithelial cells pulled from 5 pre-pregnancy female mice and 5 post-pregnancy female mice.

Sup.Fig.2c – Gene expression analysis, demonstrating increased CSN2 mRNA in mammary organoids derived from post-pregnancy MECs with and without pregnancy hormone treatment. n = 3 independent biological replicates. Each biological replicate was generated utilizing mammary organoids pulled from 2 pre-pregnancy female mice and 2 post-pregnancy female mice.

Sup.Fig.2f – Intracellular flow cytometry analysis, demonstrating increased CSN2 levels in mammary organoids derived from post-pregnancy MECs with and without pregnancy hormone treatment. n = 3 independent biological replicates. Each biological replicate was generated utilizing mammary organoids pulled from 2 pre-pregnancy female mice and 2 post-pregnancy female mice.

Fig.3g – H&E stained mammary gland images from nulliparous CAG-only control mice, transplanted with pre-pregnancy and/or post-pregnancy CAGMYC MaSCs and treated with DOX for 30 days, demonstrating the persistence of malignant lesions in glands injected with pre-pregnancy MaSCs, and demonstrating the absence of malignant lesions in glands injected with post-pregnancy MaSCs. n=4 mammary glands injected with pre-pregnancy MaSCs, and n=4 mammary glands injected with post-pregnancy MaSCs.

Fig. 3i – Abnormal branching quantification of 3D Matrigel™ mammary organoid cultures of pre- and post-pregnancy CAGMYC MECs, grown with Essential media, with or without DOX (0.5mg/mL). n=2 independent biological replicates and 3 technical replicates per experiment. Each biological replicate was generated utilizing mammary organoids pulled from 1 CAGMYC pre-pregnancy female mouse and 1 CAGMYC post-pregnancy female mouse.

Fig. 3j - Organoid size quantification of 3D Matrigel™ mammary organoid cultures of pre- and post-pregnancy CAGMYC MECs, grown with Essential media, with or without DOX (0.5mg/mL). n=100 organoids per independent biological replicate (n=2) and 3 technical replicates per experiment. Each biological replicate was generated utilizing mammary organoids pulled from 1 CAGMYC pre-pregnancy female mouse and 1 CAGMYC post-pregnancy female mouse.

Sup.Fig. 4h – General branching quantification of 3D Matrigel™ mammary organoid culture of pre- and post-pregnancy CAGMYC MECs, grown with Essential media, with or without DOX (0.5mg/mL). n=2 independent biological replicates and 3 technical replicates per experiment. Each biological replicate was generated utilizing mammary organoids pulled from 1 CAGMYC pre-pregnancy female mouse and 1 CAGMYC post-pregnancy female mouse.

Fig.4e – Density plot showing computationally defined e-box DNA binding motifs with high ATAC-seq peak intensity in MECs harvested from CAGMYC nulliparous female mouse (DD5), compared to ATAC-seq peak intensity at same e-box DNA binding site in MECs harvested from CAGMYC parous female mouse (DD5), demonstrating decreased peak intensity in post-pregnancy CAGMYC MECs. n=2 independent biological replicates. Each biological replicate

was generated from MECs pulled from 1 CAGMYC pre-pregnancy female mouse and 1 CAGMYC post-pregnancy female mouse, after DOX treatment for 5 days.

Fig.4f - cMYC Cut&Run peak enrichment analysis, showing decreased cMYC occupancy in post-pregnancy CAGMYC MECs (DD5). Peaks for display were statistically significant, defined with a pvalue=0.05 or lower using DESeq2. n=2 independent biological replicates for pre-pregnancy CAGMYC MECs and n=3 independent biological replicates for post-pregnancy CAGMYC MECs. Each biological replicate was generated from MECs pulled from 1 CAGMYC pre-pregnancy female mouse and 1 CAGMYC post-pregnancy female mouse, after DOX treatment for 5 days.

Fig. 4g – Genome Browser tracks showing differential cMYC peaks in DOX-treated, pre- and post-pregnancy CAGMYC MECs.

Fig. 4h - Western blot of cMYC, p300, and acethyl-p300 proteins in organoid cultures derived from pre-pregnancy and post-pregnancy CAGMYC MECs, with and without DOX treatment (2 days), demonstrating decreased p300 levels in post-pregnancy CAGMYC organoid cultures.

Fig. 4i - Number of branched organoids from 3D Matrigel™ mammary organoid cultures of pre- and post-pregnancy CAGMYC MECs, grown with Essential media and DOX (2 days, 0.5mg/mL), with and without Histone Acetyltransferase inhibitors (HATi II = HAT Inhibitor II 10µM and PU139 20µM). n=30 organoids per biological replicate (n=2). Each biological replicate was generated utilizing mammary organoids pulled from 1 CAGMYC pre-pregnancy female mouse and 1 CAGMYC post-pregnancy female mouse.

Fig. 4j - Organoid size quantification of 3D Matrigel™ mammary organoid cultures of pre- and post-pregnancy CAGMYC MECs, grown with Essential media and DOX (5 days, 0.5mg/mL), with and without cMYC inhibitor (cMYCI = c-Myc Inhibitor (10058-F4) 10µM). n=30 organoids per biological replicate (n=2). Each biological replicate was generated utilizing mammary organoids pulled from 1 CAGMYC pre-pregnancy female mouse and 1 CAGMYC post-pregnancy female mouse.

Sup. Fig.5b,c - Gene expression analysis, demonstrating minimal changes to previously described parous signature in pre- and post-pregnancy CAGMYC MECs (b= luminal, c= myoepithelial) after cMYC overexpression (DOX treatment). n=2 RNA-seq biological replicates. Each biological replicate was generated utilizing mammary epithelial cells pulled from 1 pre-pregnancy female mouse and 1 post-pregnancy female mouse.

Sup. Fig.5d - Differential RNA expression (Log2Foldchange) analysis of cMYC downstream target gene expression, comparing the effects of cMYC overexpression (CAGMYC DD5) over non-transgenic MECs (WT) in pre- and post-pregnancy luminal MECs (left panel), and the effects of cMYC overexpression in pre- and post-pregnancy CAGMYC MECs (luminal and myoepithelial). This analysis confirms that cMYC overexpression strongly induces the cMYC canonical pathway in pre-pregnancy CAGMYC MECs.

Sup. Fig.5g - Venn diagram showing unique and shared ATAC-seq peaks in pre- and post-pregnancy CAGMYC MECs treated with DOX for 5 days (DD5).

Sup. Fig.5h - Gene Ontology (GO) term analysis of ATAC-seq peaks exclusive to DD5 post-pregnancy CAGMYC MECs.

Sup. Fig.6d - Gene Ontology (GO) term analysis enriched in cMYC peaks exclusive to DD5 pre-pregnancy CAGMYC MECs.

Sup. Fig.6e - Gene Ontology (GO) term analysis enriched in cMYC peaks exclusive to DD5 post-pregnancy CAGMYC MECs.

Sup. Fig.6f – Mouse phenotype term analysis enriched in cMYC peaks exclusive to DD5 post-pregnancy CAGMYC MECs.

Sup. Fig.6g - Differential expression of autophagy associated genes in luminal MECs harvested from nulliparous and parous CAGMYC female mice, DOX D5, demonstrating enrichment for such gene class in post-pregnancy CAGMYC MECs.

Sup. Fig.6h - Differential expression of senescence associated genes in luminal MECs harvested from nulliparous and parous CAGMYC female mice, DOX D5, demonstrating enrichment for such gene class in post-pregnancy CAGMYC MECs.

Sup. Fig.6i - Western blot demonstrating STAT3 and p53 protein levels in organoid cultures derived from pre-pregnancy and post-pregnancy CAGMYC MECs, with and without DOX treatment (2 days).

Sup. Fig.6j - Representative images from 3D Matrigel™ mammary organoid cultures of pre- and post-pregnancy CAGMYC MECs, grown with Essential media and DOX (2 days, 0.5mg/mL), with and without Histone Acetyltransferase inhibitors (HATi II, 10μM, and PU139, 20μM).

Sup. Fig.6k - Representative images from 3D Matrigel™ mammary organoid cultures of pre- and post-pregnancy CAGMYC MECs, grown with Essential media and DOX (5 days, 0.5mg/mL), with and without cMYC inhibitor (cMYCi, 10058-F4, 10μM).

Table S1 - cMYC Cut&Run Transcription Factor DNA motif analysis, demonstrating enrichment for DNA motifs that are recognized by cMYC

REVIEWERS' COMMENTS:

Reviewer #3 (Remarks to the Author):

Thank you for the many details that have been added in this revision. Again, I want to state that this is a very interesting manuscript with compelling and novel points made. However, the strength in those points is weak in some respects, but can be made strong in two ways.

1. Document the impact of the results in the text instead of using non-measurable adjectives in your writing. For instance, instead of using terms like substantial, robust, abundant, reduced, rapid, notably, high, etc., why can't you state a 2-3 fold increase, a 50% change in this or that, a statistically significant increase in this outcome, or a 30% reduction in sites??
2. Histopathology and controls in your transplant work. The core finding is that cMYC overexpression does not induce lesions in mammary epithelium of post-preg mice. In response to point 5, you state that you show "sustained development of malignant lesions in MG injected with pre-preg MaSCs". No, you don't. There is no pathology to confirm a malignant lesion. It looks like you may have those, but proof and diagnosis by a licensed pathologist is needed on the slides to prove that. Also, assessing duct numbers (and abnormal ducts like in Fig 3) in sectioned tissue is complicated, and must be defined (how did you not count the same long duct more than once in a section?). Finally, it is critical that you explain the controls used that definitively show that the lack of response in the transplanted recipient animals (CAG only) were from the transplanted cells (CAGMYC cells of pre-preg) and not residual tissue from the recipient animal remaining after clearance of the fat pad. Neither of those cell types would respond to the DOX - do you have whole mounts or pictures of intact gland documenting the outgrowth from the point of injection? That would prove it.

I have several other areas of the paper that could use your attention:

1. abstract needs some data. Please add detail - the last sentence was already known. What is novel herein??
2. L53 suggesting - suggest
3. L110-119 please add specifics - enriched how many fold, what % was genic and intergenic
4. L133 and L137 - what kind of analyses? L137- what does more abundantly expressed mean? Also, L144- what percent similarity is there between the PIEs in the parous and non-parous or parous and hormone treated parous? What is novel?
5. L153 - how much overlap is there between luminal MECs and CD1d+MaSCs (what fraction of the total is the CD1d+ population)? What was the transplant fail rate and how was that taken into consideration? What were the controls that provide some confidence that transplanted ducts are what is present in the histology shown?
6. L158- Add (study design in Supp Fig 2a) - as stated, it is not clear what this figure refers to (histology of MGs). Then add "Dissociated and flow-separated mammary tissue transplanted with either..."
7. Please describe the ductal structure counting and how you were certain it was from transplant and not recipient.
8. L167 - robust=what? L180 - more rapidly - than what and where is the temporal data?
9. L200-211. Please engage a pathologist with knowledge of mammary histopath. Expansion and flattening of ducts, malignant lesion, and hyperplasia need confirming - also please ask them to evaluate vascularity and/or inflammation, which may be additional targets, based on your pictures. L211 - do you mean in mice? (ref)
10. L221-223. There are no methods describing this gene set comparison. Where were gene sets procured from, what software was used to compare them, and what was the question in doing this? Could you please plot the data using overlapping circles or add some more detail to the current figure 2d, as it is difficult to see the detail. Supp Fig 3b could be dropped as it is not clear what you are trying to show there.
11. L228- ER-a staining needs to be of much higher magnification to confirm nuclear staining. Please add an inset or additional picture to confirm.
12. L233-235 - what percent of the H3K27ac peaks were shared between DD2 and DD5 subsets?

May give strength to your statement.

13. L249 - name the targets. There is no Supp Fig 3f - I am pretty sure you mean Fig 2g. and please look at the last part of that sentence - something is awry (I don't know the point). Suggest deleting part after (Fig 2g).

14. Line 261 - seems it should be Fig 3a and Supp Fig 4b only.

15. Many parts of L264-280 need substantiating. Please look this section over carefully, define "abnormal", show as percent of total, and please show Supp Fig 4d similar to others - like Supp Fig 2 b.

16. L288 - slightly higher incidence of "normal" branching - means what? Supp Fig 4h should be shown as percent of control or removed as it does not match the pictures shown (Fig 3H).

17. L308-L320 need some details. "...a gene signature for parity in luminal MECs (Fig) revealed that both lum and myoep MECs of CAGMYC mice upregulated....". L312-320 is not convincing. Maybe it is the way the data is plotted - gene expression include 0 and part of the data is not described (lum vs myoep).

18. L323-328 - notably alter, expanded the number, markedly reduced - please do some stats on these data and state the details of differences.

L329 - Do you mean Supp Fig 5e or 5f? high # of PIEs - details.

L350 - Supp Fig 6a only shows EphA2, so might want to move that call out - as Tbx3 increases after DD2 in Supp Fig 3f.

19. L 405-407 - as shown, you cannot say that, but if you do the correct stats analysis, you may be able to .. you are not making the comparison in current stats shown.

Point-by-point response to the Reviewer's comments

RE: NATURE COMMUNICATION - NCOMMS-18-30590B-Z.

We thank the Reviewer #3 for the feedback and the constructive guidance to improve our manuscript entitled "Pregnancy reprograms the epigenome of mammary epithelial cells and blocks the development of premalignant lesions in response to cMYC-overexpression". We believe we have sufficiently addressed all raised concerns. Below, please find below a detailed point-by-point responses (in blue) to all of the comments from the Reviewer #3 (in black).

Reviewer #3

Thank you for the many details that have been added in this revision. Again, I want to state that this is a very interesting manuscript with compelling and novel points made. However, the strength in those points is weak in some respects, but can be made strong in two ways.

Major point 1. Document the impact of the results in the text instead of using non-measurable adjectives in your writing. For instance, instead of using terms like substantial, robust, abundant, reduced, rapid, notably, high, etc., why cant you state a 2-3 fold increase, a 50% change in this or that, a statistically significant increase in this outcome, or a 30% reduction in sites??

Answer to Major point 1. We have gone through the entire manuscript and included, wherever possible, a quantitative description of the results, as suggested by the reviewer.

Major point 2a. Histopathology and controls in your transplant work. The core finding is that cMYC overexpression does not induce lesions in mammary epithelium of post-preg mice. In response to point 5, you state that you show "sustained development of malignant lesions in MG injected with pre-preg MaSCs". No, you don't. There is no pathology to confirm a malignant lesion. It looks like you may have those, but proof and diagnosis by a licensed pathologist is needed on the slides to prove that.

Answer to Major point 2a. 9. Dr. John Erby Wikinson (University of Michigan) is an experienced mouse pathologist and a co-author in our manuscript. He has reviewed all of the histology results presented in our manuscript, which was vital to the main conclusions of the study. To ensure clarification of these results, we have now modified the results sections and figure legends to employ his diagnosis of each tissue analyzed in our study, as follows:

Fig.2a, right panel – moderate (DD2) to severe, diffuse (DD5) epithelial hyperplasia with atypia.

Fig.3a, left panel - severe diffuse, epithelial hyperplasia with atypia.

Fig.3b – right panel – normal-like tissue histology

Fig.3d – left panel – severe complex epithelial hyperplasia with atypia.

Fig.3d – right panel – normal-like tissue histology

Fig.3g – upper panel – undifferentiated carcinoma

Fig.3g – lower panel - normal-like tissue histology

Major point 2b. Also, assessing duct numbers (and abnormal ducts like in Fig 3) in sectioned tissue is complicated, and must be defined (how did you not count the same long duct more than once in a section?).

Answer to Major point 2b. For ductal quantification, mammary gland H&E histological images were uploaded into Image J. From this high-resolution image, only ducts present in the posterior part of the gland (half) were manually counted. This strategy was chosen to decrease the chance of long ducts being counted several times.

Major point 3. Finally, it is critical that you explain the controls used that definitively show that the lack of response in the transplanted recipient animals (CAG only) were from the transplanted cells (CAGMYC cells of pre-preg) and not residual tissue from the recipient animal remaining after clearance of the fat pad. Neither of those cell types would respond to the DOX - do you have whole mounts or pictures of intact gland documenting the outgrowth from the point of injection? That would prove it.

Answer to Major point 3. As pointed out by the reviewer, endogenous mammary tissue from CAG only female mice would not respond to DOX treatment, thus retaining endogenous cMYC protein levels, which were mostly undetectable by western blot and IF imaging (Fig.2 and Fig.3). Thus, for the experiments shown on Fig.3d, high levels of cMYC protein after DOX treatment (IF, Fig.3f) was utilized to confirm that transplanted mammary glands were reconstituted with pre- or post-pregnancy CAGMYC. It is also important to note that for all transplantation assays, we utilized pre- or post-pregnancy, CAGMYC, CD1d+ MaSCs, a population of stem-like cells with a mammary reconstitution success rate of 100% per 25 or more injected cells (dos Santos et al 2013 and Frey et al 2017). In the current study, we utilized 100K CAGMYC, CD1d+ MaSCs per transplant, thus allowing for the high mammary reconstitution success rate.

I have several other areas of the paper that could use your attention:

Point 1. abstract needs some data. Please add detail - the last sentence was already known. What is novel herein??

Answer to Point 1. We have rewritten the abstract, as follows:

Pregnancy causes a series of cellular and molecular changes in mammary epithelial cells (MECs) of female adults. In addition, pregnancy can also modify the predisposition of rodent and human MECs to initiate oncogenesis. Here, we investigated how pregnancy reprograms enhancer chromatin in the mammary epithelium and influences the transcriptional output of the oncogenic transcription factor cMYC. We found that pregnancy induces an expansion of the active cis-regulatory landscape of MECs, which influenced the activation of pregnancy-related programs during re-exposure to pregnancy hormones *in vivo* and *in vitro*. Using inducible cMYC overexpression, we demonstrated that post-pregnancy MECs are resistant to the downstream molecular programs induced by cMYC, a response that blunted carcinoma initiation, but did not perturb the normal pregnancy-induced epigenomic landscape. cMYC overexpression drove post-pregnancy MECs into a senescence-like state, and perturbations of this state increased malignant phenotypic changes. Taken together, our findings provide further insight into the cell-autonomous signals in post-pregnancy MECs that underpin the regulation of gene expression, cellular activation, and resistance to malignant development.

Point 2. L53 suggesting – suggest

Answer to Point 2. We have fixed this typo.

Point 3. L110-119 please add specifics - enriched how many fold, what % was genic and intergenic

Answer to Point 3. We have included more information to the manuscript text as follows (*italic blue font*).

To determine whether this response to re-exposure to pregnancy signals was linked to epigenetic changes, we profiled the active histone mark H3K27ac in the same cohort of luminal MECs subjected to RNA-seq. Total peak analysis revealed that pregnancy substantially expanded the active regulatory landscape of luminal MECs, with post-pregnancy MECs displaying a ~10-fold increase in H3K27ac peaks (n=207,585), in contrast to pre-pregnant MECs (n=19,985) (Fig. 1c). Regulatory regions exclusive to post-pregnancy MECs showed a 38-fold gain of H3K27ac peaks at genic regions (n=145,917), and a 53-fold gain at intergenic regions (n=45,174), over the same regions in pre-pregnancy MECs, suggesting that pregnancy-induced changes may expand the MEC enhancer landscape (Supplementary Fig. 1c).

Point 4. L133 and L137 - what kind of analyses? L137- what does more abundantly expressed mean? Also, L144- what percent similarity is there between the PIEs in the parous and non-parous or parous and hormone treated parous? What is novel?

Answer to Point 4a. We have included more information to the manuscript text as follows:

H3K27ac peaks exclusive to pre-pregnancy MECs define ~5,000 enhancers/super enhancers, in contrast to ~60,000 enhancers/super enhancer defined by H3K27ac peaks exclusive to post-pregnancy MECs (Parity Induced Elements, PIEs), consistent with pregnancy expanding the active enhancer landscape (Supplementary Fig.1g). Investigator on PEIS during EPH demonstrated that most PIEs have the H3K27ac mark in MECs harvested from both first and second EPH (n=2,263), however more PIEs (2-fold) were active only during the second EPH, suggesting that such elements play a role during re-exposure to pregnancy hormones (Supplementary Fig. 1h). Furthermore, we identified ~15K genes associated with PIEs, which we used to understand the effects of exposure to pregnancy hormones and gene reactivation of luminal MECs. Over 600 PIE-associated genes were up-regulated 16-fold or higher in luminal MECs harvested from mice during a second EPH (Log2FoldChange >4, red box), in comparison to those cells harvested from mice exposed during first EPH (Fig. 1e). These upregulated PIE genes were enriched for functions involved in milk production [26], thus supporting that the pregnancy-induced enhancer landscape associates with activation of pregnancy-related programs in response to re-exposure to pregnancy hormones.

Point 5. L153 - how much overlap is there between luminal MECs and CD1d+MaSCs (what fraction of the total is the CD1d+ population)? What was the transplant fail rate and how was that taken into consideration? What were the controls that provide some confidence that transplanted ducts are what is present in the histology shown?

Answer to Point 5. CD1d+ MaSCs represent ~1% of the total population of CD24⁺CD29^{high} mammary myoepithelial cells. CD1d+ MaSCs are not detected within luminal MECs by flow cytometry, but have been shown to give rise to both myoepithelial and luminal MECs in fat pad transplantation assays (dos Santos et al 2013). As mentioned above, our previous studies demonstrated a mammary reconstitution success rate of 100% when injecting 25 or more CD1d+ MaSCs (dos Santos et al 2013 and Frey et al 2017). More specifically, the mammary fatpad transplantation experiments referred to on L153 (Fig.1), were performed utilizing ~1K pre- or post-pregnancy CD1d+ MaSCs, thus within the cell range of high mammary reconstitution

success rate. Non-injected, cleared fatpads were utilized as controls for endogenous tissue removal. It is also important to highlight that the results obtained from mammary gland transplants were also reflected in organoid cultures, thus suggesting the robustness of the effects described on this manuscript, in an assay-independent manner.

Point 6. L158- Add (study design in Supp Fig 2a) - as stated, it is not clear what this figure refers to (histology of MGs). Then add "Dissociated and flow-separated mammary tissue transplanted with either..."

Answer to Point 6. We have included more information to the manuscript text as follows:

Cleared fatpads from pre-pubescent, virgin female mice were transplanted with either pre- or post-pregnancy CD1d+ MaSCs, which have increased mammary reconstitution activity in fatpad transplants [27]. Recipient female mice (2-months post-transplantation) were exposed to pregnancy hormones for 6 days, followed by histological analysis of their mammary glands (Supplementary Fig. 2a). Dissociated and flow cytometer analyzed mammary tissue transplanted with either pre- or post-pregnancy MaSCs showed comparable ratios of luminal and myoepithelial cells after tissue engraftment, suggesting that pregnancy did not affect lineage commitment and differentiation in transplanted MECs (Supplementary Fig. 2b).

Point 7. Please describe the ductal structure counting and how you were certain it was from transplant and not recipient.

Answer to Point 7. For ductal quantification, mammary gland H&E histological images were uploaded into Image J, and ducts present in the posterior part of the gland (half) were manually counted. This strategy was chosen to decrease the chance of long ducts being counted several times. Non-injected, cleared fatpads were utilized as controls for endogenous tissue removal.

Point 8. L167 - robust=what? L180 - more rapidly - than what and where is the temporal data?

Answer to Point 8. We have included more information to the manuscript text as follows:

Enhanced branching morphogenesis in response to re-exposure to pregnancy hormones was also recapitulated in *in vitro* cultures of murine mammary organoids. Post-pregnancy mammary organoid cultured with estrogen, progesterone, and prolactin hormones (complete medium) displayed a 2.3-fold greater number of branching organoids compared to pre-pregnancy cultures (Fig. 1h,i). Furthermore, additional analysis demonstrated increased *Csn2* mRNA levels (10-fold) and increased CSN2 protein levels (~4-fold), in post-pregnancy organoids cultured with pregnancy hormones compared to pre-pregnancy organoids grown under the same hormone conditions (Fig. 1j, Supplementary Fig. 2c-d, and Supplementary Table 1). Given that *Csn2* was amongst the genes elevated during second EPH (Fig.1e), our results support that cell-autonomous signals control phenotypic and molecular alterations in response to re-exposure pregnancy hormones.

Point 9. L200-211. Please engage a pathologist with knowledge of mammary histopath. Expansion and flattening of ducts, malignant lesion, and hyperplasia need confirming - also please ask them to evaluate vascularity and/or inflammation, which may be additional targets, based on your pictures. L211 - do you mean in mice? (ref)

Answer to Point 9. To address this issue, Dr. John Erby Wilkinson has revised his pathological assessment of all images. He confirmed that cMYC overexpression induces various level of epithelial hyperplasia in mammary glands from pre-pregnancy mice, in addition to confirming the normal-like state of glands from post-pregnancy female mice. Dr. Wilkinson did not find obvious alterations to vascularity or signs of inflammation.

We have included a more detailed description of tissue analysis to the manuscript text as follows, in addition to provide a more relevant reference to L211.

Thus, to investigate cMYC-driven oncogenesis in live, healthy animals, we analyzed mammary glands from CAGMYC female mice after 2 (DD2) or 5 days (DD5) of DOX treatment. Two- or 5-day DOX treatments induced substantial histo-pathological alteration to the mammary gland, including flattening of ductal structures and moderate (DD2) to severe, diffuse (DD5) epithelial hyperplasia with atypia, alterations frequently observed in premalignant mammary lesions in mice [32] (Fig. 2a, right panels). None of these alterations were seen in the control CAG-only transgenic mice (Fig. 2a, left panel). Analysis of cytokeratin composition in CAGMYC female mice revealed a progressive expansion of cytokeratin 8 (KRT8) expressing cells, a hallmark of luminal-like cells [33], over the course of the DOX treatment (Fig. 2b). This phenotype was accompanied by the progressive thinning of the basal-like cells (cytokeratin 5, KRT5), often observed during mammary tissue hyperplasia (Fig. 2b).

New ref #32 - Cardiff, R.D., et al., *The mammary pathology of genetically engineered mice: the consensus report and recommendations from the Annapolis meeting*. *Oncogene*, 2000. **19**(8): p. 968-88.

Point 10. L221-223. There are no methods describing this gene set comparison. Where were gene sets procured from, what software was used to compare them, and what was the question in doing this? Could you please plot the data using overlapping circles or add some more detail to the current figure 2d, as it is difficult to see the detail. Supp Fig 3b could be dropped as it is not clear what you are trying to show there.

Answer to Point 10. The analysis referred to here (comparing the CAGMYC transcriptome to additional publicly available mammary tumor transcriptomes), was requested by Reviewer #2 on the first round of revisions. This analysis provided additional evidence to characterize how gene expression changes present in MECs from the CAGMYC model compared to those from other models of mammary oncogenesis. The transcriptome profiles utilized in such analysis were previously curated and published, and indicated in our manuscript as reference #34 : Hollern, D.P., M.R. Swiatnicki, and E.R. Andrechek, *Histological subtypes of mouse mammary tumors reveal conserved relationships to human cancers*. *PLoS Genet*, 2018. **14**(1): p. e1007135. A brief description of the data analysis was present in the previous draft of Supplementary Information, RNA-seq library preparation and analysis, L160-162. Such analysis cannot be plotted with venn diagram plots (overlapping circles), given that it would not consider all expressed genes, in addition to variation in their level of expression. Principal Component analysis however is a more appropriate way to demonstrate variation of all genes across the samples, and to show how the transcriptome of CAGMYC MECs compared to those of other models of mammary oncogenesis.

To help clarify this data analysis, we have expanded the methods description in the Supplementary Information:

Mammary tumor tissue gene expression analysis. For the analysis presented on Fig.2d and Supplementary Fig.3b, a total of 157 mammary tumor tissue samples were retrieved from

publicly available datasets (GEO numbers GSE13221, GSE15904,GSE30805), and preprocessed using the 'affy' R package [12], in order to obtain gene expression levels. Batch effect normalization of CAGMYC RNA-seq and publicly available microarray data was performed using the ComBat function from the 'sva' R package [13]. Principal component analysis was done using the base R principle component function in order to project all gene expression data into a low dimensional space.”

Point 11. L228- ER-a staining needs to be of much higher magnification to confirm nuclear staining. Please add an inset or additional picture to confirm.

Answer to Point 11. We have included a DAPI/ER α signal overlay alone, and a higher magnification image to Supp. Fig.3c, with indications of areas where ER α signal overlaps with DAPI (* malignant lesion), and areas where ER α signal does not overlap with DAPI (^, normal duct structures), to address the reviewer’s request.

Point 12. L233-235 - what percent of the H3K27ac peaks were shared between DD2 and DD5 subsets? May give strength to your statement.

Answer to Point 12. We have included more information in the manuscript text as follows:

To investigate the effects of short-term *cMYC* overexpression on the epigenome of MECs we mapped the active enhancer landscape (H3K27ac ChIP-seq) of total CAGMYC MECs. Many (96%) of the H3K27ac peaks present in DD5 CAGMYC MECs were also present in DD2 CAGMYC MECs, suggesting that development of premalignant mammary lesions largely rely on programs activated during the initial response to *cMYC* overexpression (Supplementary Fig. 3d,e).

Point 13. L249 - name the targets. There is no Supp Fig 3f - I am pretty sure you mean Fig 2g. and please look at the last part of that sentence - something is awry (I don't know the point). Suggest deleting part after (Fig 2g).

Answer to Point 13. We have included more information to the manuscript text as follows:

Genome browser tracks illustrate increased H3K27ac levels in MECs after induction of *cMYC* overexpression, at *cMYC* downstream targets *Tbx3* and *Reep5*, both of which have been implicated in mammary oncogenesis [37, 38] (Fig. 2g). Thus, short-term *cMYC* overexpression activates specific epigenomic and transcriptional networks, and causes alterations to tissue morphology resembling those of murine mammary oncogenesis.

Point 14. Line 261 - seems it should be Fig 3a and Supp Fig 4b only.

Answer to Point 14. We have altered the sentence as follows to justify the mentioning of main and supplemental figures as originally pointed.

Histological analysis revealed that mammary glands of nulliparous female mice displayed a ductal content 3-fold higher than mammary glands from parous CAGMYC female mice, which remained largely unaffected by *cMYC* overexpression (Fig.3,ab). In agreement, mammary glands from parous CAGMYC female mice showed tissue morphology and duct numbers (276 \pm 42) similar to those from the DOX-treated, CAG-only control group (382 \pm 4), supporting that post-pregnancy mammary glands retained a mostly normal phenotype in response to *cMYC*

overexpression (Supplementary Fig. 4b,c). These phenotypic differences were not caused by inefficient transgene induction, as pre- and post-pregnancy CAGMYC MECs expressed comparable *cMYC* mRNA and protein levels (Fig. 3c, Supplementary Fig. 4d).

Point 15. Many parts of L264-280 need substantiating. Please look this section over carefully, define "abnormal", show as percent of total, and please show Supp Fig 4d similar to others - like Supp Fig 2 b.

Answer to Point 15. We have altered the sentence as follows to address the reviewer request:

In response to *cMYC* overexpression, mammary fatpads transplanted with pre-pregnancy CAGMYC MECs demonstrated severe complex epithelial hyperplasia with atypia and abnormal ductal morphology, in contrast to fatpads transplanted with post-pregnancy CAGMYC MECs, which displayed mostly normal tissue histology and lacked abnormal ductal structures (Fig. 3d, e). There were no significant changes on the total number of ducts, or *cMYC* protein levels, in glands transplanted with either pre-pregnancy (217 \pm 49 ducts) or post-pregnancy (140 \pm 9 ducts) CAGMYC MECs, suggesting that the lack of abnormal ductal clusters in the post-pregnancy condition was not an artifact associated with transplantation of *cMYC* overexpressing cells (Fig. 3e,f). Extending *cMYC* overexpression to 30 days (DD30) also failed to induce the development of premalignant lesions in mammary glands transplanted with post-pregnancy CAGMYC CD1d+ MaSCs, in contrast to glands transplanted with pre-pregnancy CAGMYC CD1d+ MaSCs, which progressed from epithelial hyperplasia to undifferentiated carcinoma lesions (Fig. 3g). Collectively, these results are consistent with *cMYC* overexpression being less efficient at driving malignant transformation of post-pregnancy MECs.

Point 16. L288 - slightly higher incidence of "normal" branching - means what? Supp Fig 4h should be shown as percent of control or removed as it does not match the pictures shown (Fig 3H).

Answer to Point 16. We have altered the sentence as follows to address the reviewer's request:

Analysis of pre- and post-pregnancy CAGMYC organoid cultures exposed to increasing concentrations of DOX demonstrated similar induction of *cMYC* protein levels (Supplementary Fig. 4f, g). Morphological analysis of untreated organoid cultures revealed that pre- and post-pregnancy CAGMYC organoid cultures displayed similar morphology, with pre-pregnancy organoids displaying a higher incidence of normal branching (39 organoids, 35% of total organoids) than post-pregnancy organoids (10 organoids, 8% of total organoids), possibly due to differences on cell culture adaptation (Fig. 3h - left panel, Supplementary Fig. 4h).

Point 17. L308-L320 need some details. "...a gene signature for parity in luminal MECs (Fig) revealed that both lum and myoep MECs of CAGMYC mice upregulated....". L312-320 is not convincing. Maybe it is the way the data is plotted - gene expression include 0 and part of the data is not described (lum vs myoep).

Answer to Point 17. Supplementary Figures 5b and 5c show the Log2FoldChange values from differentially expressed genes comparing pre and post-pregnancy, luminal and myoepithelial,

CAGMYC MECs, a common way to display changes to expression when comparing two samples. A similar analysis, comparing genes differentially expressed in wild-type luminal pre and post-pregnancy MECs is demonstrated on Supplementary Fig. 1b. Specific to this analysis, a Log2FoldChange values greater than 2 (4-fold difference on mRNA levels) were considered to call a gene upregulated in post-pregnancy MECs. To improve data plot interpretation, we positioned the red arrow to indicate Log2FoldChange values greater than 2, and blue arrow to indicate Log2FoldChange values lower than 2, and explained this positioning on figure legend. We have also made alterations to results session to include more information as follows:

a) Analysis of wild-type luminal pre and post-pregnancy MECs, demonstrated on Supplementary Fig. 1b:

Focused analysis of 47 genes correlated with MEC parity status [25] confirmed the upregulation of 38% of the parity-induced genes in post-pregnancy luminal MECs (Supplementary Fig. 1b).

b) Analysis of pre and post-pregnancy CAGMYC MECs, demonstrated on Supplementary Fig. 5b and 5c:

Analysis of the 47 parity-associated factors (Supplementary Fig.1b) showed that 19% and 23% of these gene signatures remain upregulated in post-pregnancy luminal and myoepithelial CAGMYC MECs, respectively, suggesting that pregnancy-associated transcription signatures are not substantially altered by *cMYC* overexpression (Supplementary Fig. 5b, c).

Point 18. L323-328 - notably alter, expanded the number, markedly reduced - please do some stats on these data and state the details of differences. L329 - Do you mean Supp Fig 5e or 5f? high # of PIEs - details. L350 - Supp Fig 6a only shows *Epha2*, so might want to move that call out - as *Tbx3* increases after DD2 in Supp Fig 3f.

Answer to Point 18. We have altered the sentences as follows to address the reviewer request:

a) L329 - Do you mean Supp Fig 5e or 5f? high # of PIEs – details:

cMYC overexpression induced a 6-fold increase in H3K27ac signal intensity at promoter regions in pre-pregnancy CAGMYC MECs, compared to promoter regions from pre-pregnancy WT MECs (Fig.4c). Conversely, the effect of *cMYC* overexpression on promoter regions was not as strong in post-pregnancy CAGMYC MECs, which displayed 3-fold less H3K27ac signal intensity compared to those of pre-pregnancy CAGMYC MECs (Fig.4c). This differential response to *cMYC* overexpression was also reflected on the total number of detected H3K27ac peaks, with 26% (n=890) mapping to promoter regions in pre-pregnancy, compared to 6% (n=295) of promoter regions in post-pregnancy CAGMYC MECs (Supplementary Fig. 5e). Conversely, a larger percentage of H3K27ac peaks from post-pregnancy CAGMYC MECs mapped to genic regions (80%), compared to pre-pregnancy CAGMYC MECs (60%), consistent with pregnancy inducing an expansion of putative cis-regulatory regions in MECs (Fig.1), which was not significantly altered by *cMYC* overexpression (Supplementary Fig. 5e). Analysis of H3K27ac intensity levels at PIEs demonstrated retained, parity-induced high H3K27ac levels in post-pregnancy CAGMYC MECs, indicating *cMYC* overexpression did not perturb pregnancy-induced epigenomics signatures (Fig.4d).

b) L350 - Supp Fig 6a only shows Epha2, so might want to move that call out - as Tbx3 increases after DD2 in Supp Fig 3f:

In order to compare changes in H3K27ac levels with cMYC DNA occupancy we analyzed the Epha2 gene, which codes for a tyrosine receptor kinase expressed in mammary tumors [39], and found it to display decreased H3K27ac intensity in response to cMYC overexpression in post-pregnancy CAGMYC MECs (Supplementary Fig. 6a). ChIP-qPCR of Epha2 and Tbx3 genes (Supplementary Fig. 3f) revealed a ~3-fold higher cMYC DNA occupancy in pre-pregnancy CAGMYC MECs, further supporting that cMYC is less efficient at associating with chromatin at these genomic regions in post-pregnancy CAGMYC MECs (Supplementary Table 2, Supplementary Fig. 6b,c).

Point 19. L 405-407 - as shown, you cannot say that, but if you do the correct stats analysis, you may be able to .. you are not making the comparison in current stats shown.

Answer to Point 19. We have altered the sentences as follows to reflect the analysis presented on Fig.4j.

These alterations were not dependent on blocking cMYC activity, given that pre-pregnancy CAGMYC organoids remained 2.9-fold larger than organoids derived from CAGMYC post-pregnancy MECs, after treatment with cMYC inhibitor (cMYCi) (Fig. 4j, Supplementary Fig. 6k), thus supporting that specific perturbations to post-pregnancy CAGMYC MECs can revert their ability to respond to cMYC overexpression, and engage on malignant transformation.